# Use of a controlled experiment and computational models to measure the impact of sequential peer exposures on decision making

Soumajyoti Sarkar[1]*, Paulo Shakarian[1], Danielle Sanchez[2], Mika Armenta[2], Kiran Lakkaraju[2]

**1** School of Computing, Informatics, and Decision Systems Engineering, Arizona State University, Tempe, AZ, United States of America, **2** Computer Science and Applications, Sandia National Laboratories, Albuquerque, NM, United States of America

* ssarka18@asu.edu

**Data Availability Statement:** The data for our simulation of diffusion has been obtained from a

## Abstract

It is widely believed that one's peers influence product adoption behaviors. This relationship has been linked to the number of signals a decision-maker receives in a social network. But it is unclear if these same principles hold when the "pattern" by which it receives these signals vary and when peer influence is directed towards choices which are not optimal. To investigate that, we manipulate social signal exposure in an online controlled experiment using a game with human participants. Each participant in the game decides among choices with differing utilities. We observe the following: (1) even in the presence of monetary risks and previously acquired knowledge of the choices, decision-makers tend to deviate from the obvious optimal decision when their peers make a similar decision which we call the *influence decision*, (2) when the quantity of social signals vary over time, the forwarding probability of the influence decision and therefore being responsive to social influence does not necessarily correlate proportionally to the absolute quantity of signals. To better understand how these rules of peer influence could be used in modeling applications of real world diffusion and in networked environments, we use our behavioral findings to simulate spreading dynamics in real world case studies. We specifically try to see how cumulative influence plays out in the presence of user uncertainty and measure its outcome on rumor diffusion, which we model as an example of sub-optimal choice diffusion. Together, our simulation results indicate that sequential peer effects from the influence decision overcomes individual uncertainty to guide faster rumor diffusion over time. However, when the rate of diffusion is slow in the beginning, user uncertainty can have a substantial role compared to peer influence in deciding the adoption trajectory of a piece of questionable information.

publicly available dataset in https://journals.plos.org/plosone/article?id=10.1371/journal.pone.0150989.

**Funding:** Some of the authors are supported through the ARO grant W911NF-15-1-0282 and W911NF-19-1-0066. Sandia National Laboratories is a multimission laboratory managed and operated by National Technology & Engineering Solutions of Sandia, LLC, a wholly owned subsidiary of Honeywell International Inc., for the U.S. Department of Energy's National Nuclear Security Administration under contract DE-NA0003525. This paper describes objective technical results and analysis. Any subjective views or opinions that might be expressed in the paper do not necessarily represent the views of the U.S. Department of Energy or the United States Government. The funder provided support in the form of salaries for authors Kiran Lakkaraju, Paulo Shakarian, Soumajyoti Sakar, Mika Amenta, Danielle Sanchez but did not have any additional role in the study design, data collection and analysis, decision to publish, or preparation of the manuscript. The specific roles of these authors are articulated in the 'author contributions' section.

**Competing interests:** The authors declare no competing interests. The commercial affiliation to Sandia National Laboratories does not alter our adherence to PLOS ONE policies on sharing data and materials.

# 1 Introduction

The connections and interactions of individuals that comprise social networks are generally believed to impact decision-making in many domains including product selection and decision making in uncertain environments [1, 2]. While there is general theoretical consensus that social influence, the phenomenon by which an individual's opinions, behaviors, and decisions are influenced by other people [3], facilitates product selection, the empirical literature is actually quite torn. According to individual utility models, people adopt technologies when their benefits exceed their costs [4]. Because comparing every option can be cognitively costly and time consuming, individuals employ cognitive strategies and shortcuts that reduce the number of alternatives until one superior option is left [5]. Social signals factor into the strategies because they provide a cost-efficient means of acquiring information.

Some find that social information predicts selection decisions [6] and others have reported interactions between a decision-maker's experience and/or knowledge of the product and their general susceptibility to social influence [7]. It is critical that research agendas pursue understanding of these nuances because as technologies become more sophisticated, widely used, and powerful, so does their potential to be used for harm. The recent slew of worldwide cyber-attacks is a potent example of how the selection of cyber-defense venders will affect billions of people [8]; if cyber-defense 'shoppers' are subject to 'sub-optimal' social influence, how will their decisions be affected? The choice to follow the herd may not be the best strategy when the herd chooses 'wrong'. Similarly. the widespread rise in misinformation has detrimental impacts in society and where social influence has known to be adversely impacting the decision making process leading to undesirable contagion [9, 10]. In both these situations, the choice to follow the herd may not be the obvious optimal choice and social influence can play a detrimental role. In this paper, we investigate the role of *patterns of influence (PoI)* or the manner in which an individual is repeatedly subject to the same piece of information over time by being embodied in a connected environment with other individuals, on behavior diffusion. We extend our recent work on understanding the role of PoI towards individual decision-making using a controlled experimental setup [11, 12].

We developed two sets of studies in this paper. First, we started off by developing an online controlled framework to question the longstanding notions of the magnitude of peer signals as a reliable predictive factor of behavior diffusion. To this end, we developed an experimental framework to characterize the exposure effect under multiple signals—but when the pattern of influence or PoI could be controlled (we show an example of what a "pattern" is in Fig 1)—the experimental framework allows us to measure social influence while avoiding confounding effects. It allows us to analyze how the signal proportion, when paired with its temporal treatment, impacts the selection choices of users in environments where the influence decision is not the best choice. The first study was designed to avoid network effects as confounders in our understanding of peer influence effect on behavior diffusion. Following this, in a separate second study, we attempted to test some of the rules obtained from our behavioral findings in the controlled experiment on sub-optimal choice selections in networked environments in the real world. We specifically try to measure the role of peer signals in networked environments and in the presence of user uncertainty on the diffusion of information with questionable veracity. The objective of this second work is to understand the extent to which our behavioral findings from the experimental data can be applied to observational data—to this end we simulate spreading on real networks.

For our first work, we used Amazon Mechanical Turk (AMT) to run an online, controlled decision-making game and recruited participants for the game conducted over several time steps. At each time step, participants selected one technology among six choices with differing

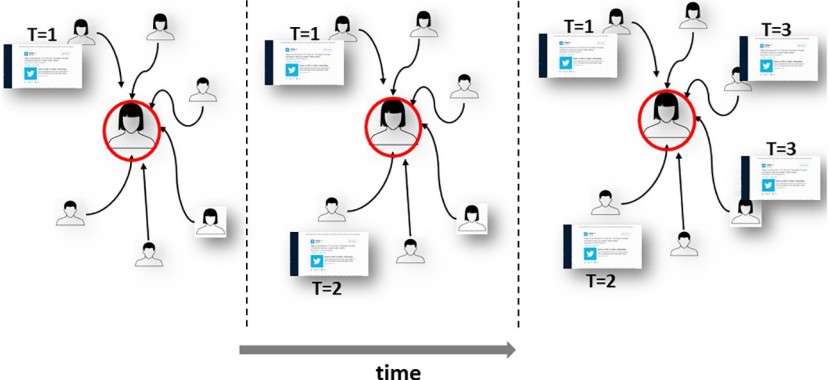

**Fig 1. This illustration demonstrates a *pattern of influence*.** At the first time step, one among six neighbors of a user shared a message, in the following time step, another user shared the same message and in the final step, two additional users. The user is thus exposed to a "pattern" of social signals we represent as $A = \{1, 2, 4\}$ for that message.

utilities (only one among them was the optimal technology)—we modulated the latter part of the game by subjecting participants to peer signals directed towards a technology which was not optimal. We focused on understanding the effects of PoI on the choices made by individuals in such a setting. While the first study relies on experimental data, in our second work we adopted a data-driven approach to simulate diffusion using multi-agent models but the agents are mapped to real world users. The diffusion model incorporated the rules of peer influence observed from the experimental setup.

Our work has been influenced from existing studies on peer exposures and behavior diffusion [13, 14], however to the best of our knowledge, the extant work does not examine the effect of signal exposure when the signals promote sub-optimal choices. In an observational study on the impact of repeated exposures on information spreading [15], the authors showed that an overwhelming majority of message samples are more probable to be forwarded under repeated exposures, compared to those under only a single exposure. However what is often understudied or left out in these research studies is the sequential exposure mechanism or what we refer to as PoI in the paper. Additionally, observing and mapping these PoI in the real world to analyze the cause and effect synopsis is not straightforward because of the opacity problem [16].

Similarly, the experiment conducted in [17] reported that individual adoption is much more likely when participants receive social reinforcement from multiple neighbors in the social network relative to a single exposure, however it did not differentiate the effects of network structure from the sequential exposure mechanism or what we refer to as a pattern of influence. One of the main results from [17] shows that the influence monotonically increases, although the increased likelihood of influence from $k$ signals compared to $k - 1$ signals peaks when $k = 2$. Our results, as summarized below, have the following observations:

- We clustered participants in the online experiment into five groups—two control groups and three treatment groups differing by the PoI or the pattern of information cascade induced among the neighbors. Upon analyzing the selections made by participants aggregated over the lifecycle of the game, we observed the following—compared to participants who were treated to a single controlled sub-optimal peer signal (reflecting the influence decision) over all time steps, the probability of selecting that *influence decision* was significantly higher for participants who were treated early to a large quantity of such controlled sub-optimal peer signals.

- Through multiple analyses on the effect of sequential peer exposures, we observed that the number of exposures alone does not explain successful social influence contrasting conclusions from several previous studies. Surprisingly, a delayed stimulus in the form of a sudden increase in peer signals is a more effective influence strategy for later stages than peer signals administered through a uniform build up when comparing the same time stage.

- Finally, as a step towards understanding how the rules from the behavioral findings play out in real world diffusion, we develop data-driven agent-based models that simulate rumor diffusion. We use data to learn agent specific parameters and evaluate the spreading dynamics of a group of questionable information in Twitter networks. We found that while social influence based on sequential exposures can result in faster diffusion compared to the factor of simply the number of peer exposures, individual behavior uncertainty can also play an important role that can impact the influence factor itself.

The rest of the paper is organized as follows: we first discuss the related literature underpinning this study in Section 2 and the hypotheses that we will investigate in Section 3. We present the experimental setup and methods designed for measuring social influence in Section 4. We analyze the controlled experiment results in Section 5. Finally, we develop an agent-based model drawing upon the conclusions of the controlled experimental results and evaluate its results with real world data in Section 6.

## 2 Related work

Informational social influence, the tendency to accept information from others as evidence about reality, tends to affect financial decision-making and product selection more than other forms of social influence [18–20]. We are more often swayed by others' decisions and behavior when we lack knowledge about the object of our decision, such as when we must choose a product that we do not know much about, is not well-described, or that we have little experience with [21]. This is because the information we seek can be more cheaply acquired by observing others than by seeking it ourselves. Conventional studies suggest that as the consensus of entities in a social network increases—more signalers make the same signal—we assume that the information peers are conveying is valid and we are more likely to adopt the signaled behavior or decision [19, 22]. The literature documents several influences on the adoption of behaviors including network structure—who is connected to who and the properties associated with these connections [17]—an individual's information parsing processes, their perceptions of product utility, and the number of signals they receive.

### Number of signals

The relationship between the number of signals an individual receives in its network, social influence, and the likelihood that said individual will adopt the behavior indicated by the signals is closely related to the linear threshold model in which an actor adopts a behavior after the signal count reaches an optimal threshold [23]. What, though, is the impact of repeated signals on the decision-making process, and more specifically how many signals are required to reliably influence an individual's decision-making? There are mixed findings regarding the benefit of multiple exposures on the diffusion of information necessary to reach this threshold [24]. People may prefer multiple confirmations from their peers to reassure themselves before making a decision [15, 25]. Experimental human studies using games from behavioral economics like the Prisoner's Dilemma tend to find that the impact of zero to three signalers increases behavioral and decision-adoption in a linear fashion, and that a key threshold for maximum social influence exists between four and five signalers. This means that two signalers

exert more influence than one, three exert more than two, and four to five exert more influence than three. There is not much difference between the impact of five and six or more. However, debate still exists—some have found the threshold to be two signalers [26] while others report it at three [17, 27]. Still, others have found the reverse trend. In one study, repeated exposure to online signals in a social networking site slowed the subsequent spread of information. This might affect decision-making and behavioral adoption by inhibiting social influence [24, 28].

## Network structure

A network structure's describes characteristics of the network as a whole. This can include properties like clustering/decentralization (to what degree actors form ties), modularity (how densely connected nodes are within clusters), homophily/heterophily (similarity or dissimilarity of actors predicts tie-forming), and centralization (how connected are actors). For instance, the density of a decision-maker's social network can influence the choices they make because signalers' intentions are more ambiguous in densely connected networks [29]. Similarly, the choices that individuals make and the contribution of social influence have also been attributed to the network communities they belong to, while studies related to patterns of subgraphs that result out of such individual interactions have been popular [30]. The dynamism, or how easily entities can move in and out of networks, also influences decision-making. Social decision-making modelers have found that the more dynamic and mobile a network, the greater concurrence of their decisions in behavioral economics games [31, 32]. Because the present studies are primarily interested in the effects of a) the number of signals and b) the pattern by which they are received, we attempt to control for the effects of network structure by holding the structure constant throughout all experimental conditions in both studies. Please see the Methods sections for further details.

## Information parsing

The manner in which individuals search for information affects their decision-making. Individuals employ search strategies to reduce the number of choices [5]. This includes revising their initial opinions by processing and averaging the different influences acting on them [33] and social information provides one mechanism through which this is achieved. These manners of decision making have also been linked to the concept of dual process theory, the notion that two different systems of thought co-exist; a quick, automatic, associative, and affective-based form of reasoning and a slow, thoughtful, deliberative process [34]. Fast thinking involves conditions of "cognitive ease" and so social influence factors into this process of slowing down the decision making system by presenting alternating evidences for reconsideration. When social signals point towards a specific outcome or opinion, individuals will often adopt the opinions and behaviors of signalers [22, 35], however while adopting behaviors based off social influence can be cost effective, it does not always lead to the most effective or efficient decision. Individuals must trade-off between trusting their own knowledge and trusting other's opinions [13]. Biased social signals can influence individuals to choose wrong or less effective answers in a variety of domains, particularly if multiple peers back the behavior or decision [15, 22, 35].

## Product utility

When making product decisions, outcomes related to the quality and need for a product change its utility or perceived value and therefore the risk of the selection. For instance, when decision-makers were asked to purchase songs in an online market where they could see the decisions of other purchasers, the perceived quality of the songs predicted their choices even

when they witnessed peers purchasing songs of poor quality. Songs of medium quality were most subject to the effects of social influence [36]. The need for a quality product also influences selection decisions. Kraut and colleagues found that when deciding between different video-teleconferencing technologies, people with the most communication intensive work— those who relied on video-teleconferencing the most—placed greater value on the product's ability than those with less need [4]. The perceived value of a product predicts how much an individual will search for information to inform their selection [21], and consumers and decision-makers are more likely to engage in information search, including social information— when purchasing products is risky—such as when the costs of making a sub-optimal choice are high, and when they lack prior knowledge about the technology [21]. When we feel that we know enough about a product, we believe that we already have enough information stored in our memory to make the best decision, and therefore additional social information is unnecessary.

## 3 The present research

We performed an online controlled experiment that manipulated the number of social signals and the signal pattern over time. We hypothesized that successful social influence requires more than just receiving signals or exposures to information from peers, as both the utility of the technology and informational influences are at play. In our experiment, any decision made produces varying degrees of monetary gain based on utility. So, we speculate that successful social influence should be reflective of the mechanism through which information diffuses and that ultimately instigates individuals to change their beliefs and therefore their adoption behavior. We present the following two hypotheses in this paper that were tested with regard to the objective mentioned above.

HYPOTHESIS H1. *Individuals will be more likely to choose a sub-optimal cyber-defense provider when they observe peers choosing the sub-optimal provider.*

Through our first hypothesis, we try to examine the aggregated results of the cascading exposure arising from varying temporal patterns of influence on the outcome of interest— whether users follow the decision made by their peers. In this hypothesis, time takes a backseat and we tried to measure the extent to which early, uniform or delayed exposure to peer influence successfully achieves our desired outcome in situations of sequential decision making when aggregated over all time steps of the experiment. Subsequently, the hypothesis attempts to examine findings in Centola's experiment [17] which show that behaviors spread to a larger portion of the population in a clustered network, indicating that additional social signals have significant effect on influence. However, the results on behavior diffusion reported by this paper are heavily associated with the clustered network organization that dictates the exposure to social influence and, as such, the sequential nature of exposures and its effect were largely ignored in the study. Following this, in another study on empirical data from Twitter [37], the authors show that the number of active neighbors is a positive indicator of influence, which is a similar finding reported by [13, 17]. In both studies, the authors did not segregate varying temporal influence patterns that might force users to revise their beliefs over time in different manners. We built on these experiments to test the aggregated effect of the peer signals and the extent to which the manner of signal dissemination among peers can act as an impetus towards coercing users to change their decisions, especially when users weigh their own private information against external influence.

HYPOTHESIS H2. *The cascading pattern of peer signals or the temporal patterns of influence will impact the adoption behavior of individuals more than just the quantity of signals.*

In this hypothesis, our goal was to understand whether the same quantity of peer signals have different outcomes when individuals are subjected to them through different cascading patterns. We again note that the authors in [17] concluded that the likelihood of a user adopting a behavior at $k$ signals compared to $k - 1$ signals was the maximum when $k = 3$. The author attributed this to the clustered network organization that allowed users to receive multiple signals before they chose to adopt a behavior. What we instead posit is that "time" has an important role to play in the revised beliefs and opinions of individuals i.e.—an early exposure to peer influence can result in a substantially higher dynamics of adoption than situations where influence is delayed. We deliberately downplay the role of networks to be able to control and study the nature of cascading and we consider that all peers of an individual are homogeneous with respect to the influence they can exert on it. This allows us to control the pattern of influence, i.e. the number of signals sent over each time step to an individual.

As we will show later, a successful social influence constitutes situations where individuals not only deviate from the optimal decision but they also select the option that majority of their peers choose. We describe the experimental setups and the formaldiscussions of our methods and conclusions in the following sections for which wesummarize the notations in Table 1.

## 4 Methods

The following protocol was approved by the Human Studies Board at Sandia National Laboratories. The consent was obtained in writing from the board. Subjects provide informed consent through written means. The "documentation of informed consent" was waived since this was an online experiment that had no more than minimal risk of harm to subjects. To test our hypotheses, we ran an online, controlled decision-making game in which participants took on the role of a security officer at a bank. Participants were told that they and several of their peers at different banks were being asked to invest in a cyber-defense technology provider once a month for 18 time steps. We separated participants into five groups based on pattern of social signal exposure which is described in details in the *Design* subsection following this. At each time step, participants were able to choose from six different technology providers—among which only one was optimal, preventing 7 attacks. The remaining five providers prevented five

**Table 1. Table of symbols.**

| Symbol | Description |
|---|---|
| $u$ | a user or an agent |
| $C_u$ | influence decision for $u$ |
| $N_u$ | number of times user $u$ has adopted the influence decision $C_u$ in $T_{treat}$ |
| $A_u(t)$ | number of peers of user $u$ who have adopted $C_u$ at time $t$ |
| $d_i$ | technology choice/decision $i$, $i \in [1, 6]$ for the controlled experiment |
| $D_u(t)$ | decision or technology adopted by user $u$ at time step $t$ |
| $n$ | number of individuals in a group in the controlled experiment |
| $T_{treat}$ | treatment phase of controlled experiment, time steps 13 to 18 in our setting |
| $G = (V, E)$ | a network $G$ consisting of nodes $V$ and edges $E$, $V(\mathcal{G})$ denotes subset of vertices relevant to network $\mathcal{G}$. |
| $p_u(t)$ | probability of user $u$ adopting a sub-optimal decision at time $t$ |
| $\mu_u(t)$ | utility obtained by user $u$ upon choosing the optimal decision at time $t$ |
| $\zeta_u(t)$ | utility obtained by user $u$ upon choosing the influence decision $C_u$ at time $t$ |
| $z_u(t)$ | binary variable that assumes value 1 if user $u$ chooses $C_u$ at time $t$, else zero |
| $\eta_u, \beta_u$ | Agent specific parameters in the ABM model |
| $q, V_q$ | information cascade and the users/nodes participating in $q$ |
| $DF_q[t]$ | Diffusion node set for cascade $q$ obtained at time step $t$ from our ABM simulation |

attacks each (from that perspective, all sub-optimal technologies had the same utility). For every attack they prevented in any time step, participants received \$0.02 (so, a participant could receive a maximum \$1.14 in each time step). Thus, they were incentivized to avoid more attacks and earn more money. This information about the optimal and sub-optimal providers is not available to the participants at the beginning of the game. Additionally, all participants could view brief descriptions of provider capabilities, e.g. one of them being "Secure.com utilizes algorithmic computer threat detection to keep systems safe. It prides itself on its efficiency and success rate in warding against attacks".

## 4.1 Design

Participants were randomly assigned to five groups with each group having unique members not involved in decision making as part of other groups. The entire game was partitioned into two phases. For the first phase comprising 12 time steps, no other information except a short excerpt about six potential providers was given. After the participants made their selection for a given time step, they saw the number of attacks their provider had prevented in that step. As mentioned earlier, in the absence of the knowledge of the utilities of the technology providers (or the number of attacks it prevents), the first 12 time steps allow for individual decision making and exploration.

In the second phase of the experiment which started at Time Step 13, we introduced interventions by allowing participants access to extra information from six other individuals who are supposedly their peers (but are really bots controlled by us). After participants make their choice at a time step, they can view the selections made by their peers in the previous time step. Each participant in all groups bar one are subjected to six peers—We call the decisions of these peers which the participant views in the second phase as peer signals in this work. Fig 2 shows the screen of a participant from time steps 13 to 18 when they were able to view the decision of their peers. The platform was hosted by the Controlled Large Online Social Experimentation (CLOSE) platform and developed at Sandia National Laboratories [38]. While the participant is able to view the technology selections of all their peers at each time step, we control social influence by administering a randomly selected sub-optimal technology $C_u$ (among the five providers) through the peers of the participant $u$—so using our nomenclature, $C_u$ is the *influence decision* for $u$. For each $u$, this technology $C_u$ was selected as the choice that is disproportionately signaled by its peers over time (this pattern of influence or PoI was manipulated by us). The motivation behind this deliberate selection of sub-optimal $C_u$ (controlled by us) as the *influence decision* was to investigate whether participants would be tempted to select this technology $C_u$ in the presence of its peer feedback. Note that we attempted to avoid network effects by randomizing this technology or the *influence decision* $C_u$ specific to the user $u$—this allows us to avoid any deliberate collisions among peer choices of different users that could be representative of network effects in the real world.

We denote by $A_u(t)$ the number of peers of the participant $u$, who we administer the sub-optimal technology/influence decision $C_u$ at time step $t$ (so $A_u(t) \leq 6$). We now describe the pattern of influence $A$ dropping the subscripts to generalize for all users or *PoI* as we denote it, for the five groups (the first two being the control groups for comparison with the next three treatment groups) (Fig 3 shows the signal patterns for the groups):

1. *No Message (NM)*: Participants receive no message from the peers, so the last six time steps are exactly the same for the participants as the first 12.

2. *Uniform Message (UM)*: Here we send $C$ using one peer of a participant at each time step. So $A = \{1, 1, 1, 1, 1, 1\}$ denotes uniform influence.

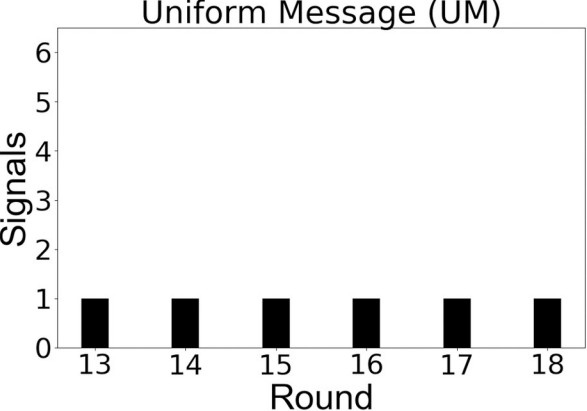

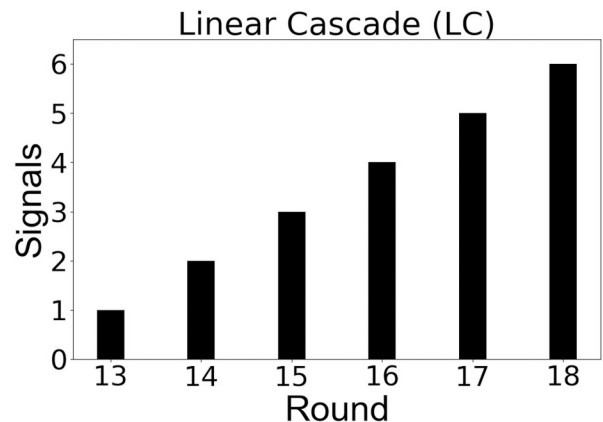

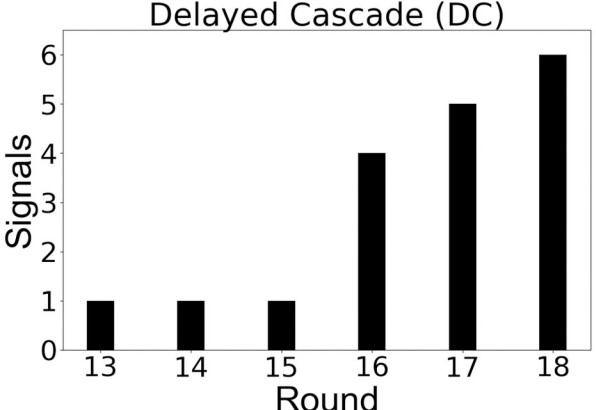

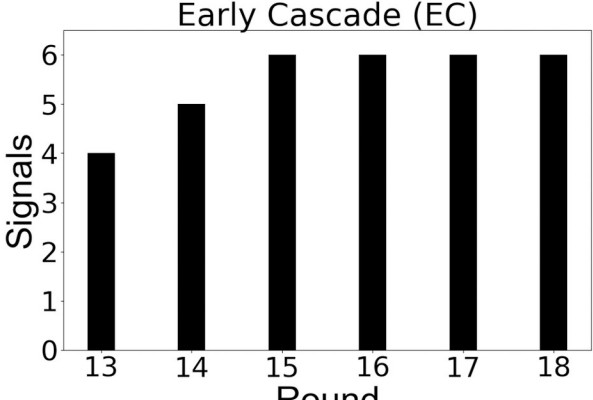

**Fig 2. Example screen in the cyber-defense provider selection task.** Participants in the Uniform Messages (UM) condition of the study have access to a screen that resembled this one. The *Feedback* section displays the number of attacks the participant prevented after each time step. The *Decisions* section displays the six provider choices that it has. Finally, the *Messages* section is displayed after Time Step 12, where the participants can view what their peers selected in the previous time step.

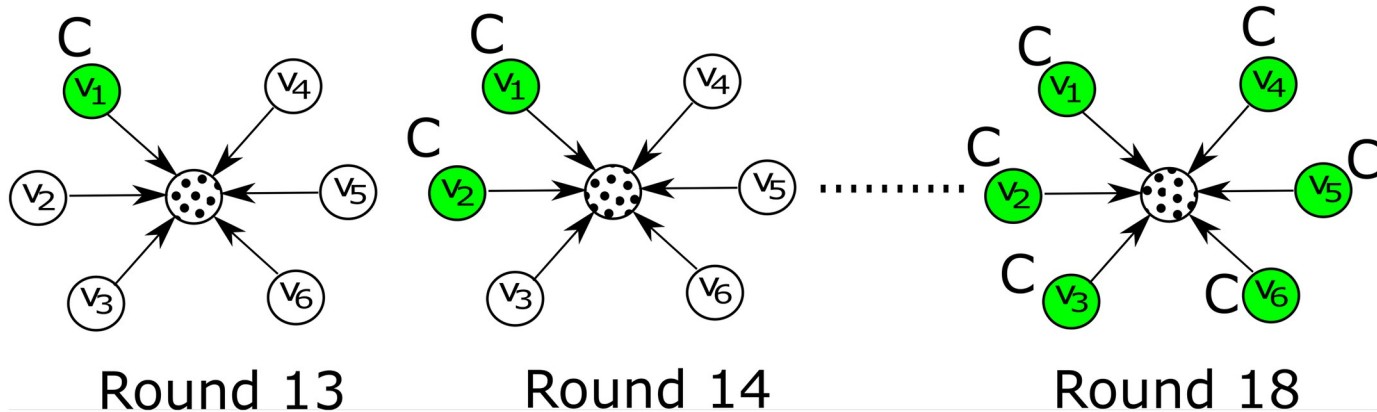

**Fig 3. The signal vs time step plots for the 4 patterns—Note that for the NM group (not shown here), no peer signal in the form of pre-selected sub-optimal technologies were sent to the participants at any time step.**

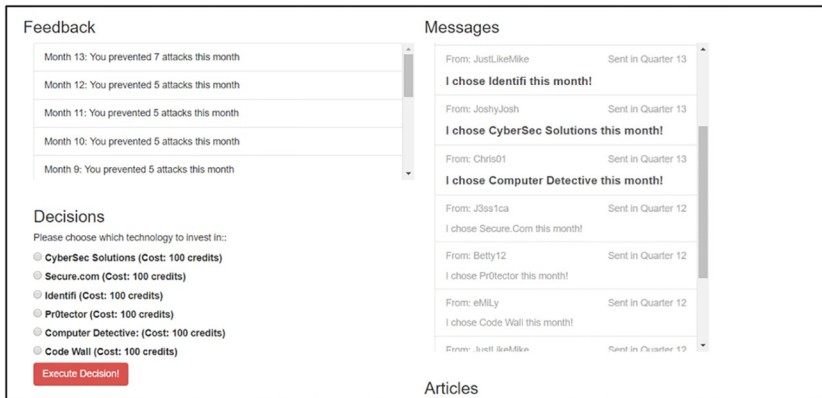

**Fig 4. Illustration of the *linear cascade* diffusion.** The technology $C_u$ chosen by us as the sub-optimal technology (*influence decision* for user $u$ (in dots) cascades through the peers of $u$ over the six time steps. Colored nodes denote the activated peers with respect to $C_u$ (manually preprogrammed by us) at each time step. Note that although at time steps starting at 13 and ending at 18, there are subjects (uncolored) among peers who have not adopted $C_u$, their selections (which may not be $C_u$) are visible to $u$. However, which users among the peers have been preprogrammed manually is by default unknown to the target subject $u$.

3. *Linear Cascade (LC)*: Here we incrementally activate one peer with the technology $C$ at each time step. So $A = \{1, 2, 3, 4, 5, 6\}$ denotes uniformly increasing influence as shown in Fig 4.

4. *Delayed Cascade (DC)*: Here we send only one signal for the first three time steps and send four, five and six signals at the last three time steps in order. So $A = \{1, 1, 1, 4, 5, 6\}$. The objective is to see whether the sudden change in the number of signals acts as a catalyst for successful influence at the later stages of the experiment.

5. *Early Cascade (EC)*: In this setup, we send a higher magnitude of signals from the beginning setting $A = \{4, 5, 6, 6, 6, 6\}$. This pattern allows us to ask if an early trigger is able to sustain the levels of influence, or whether participants will return to the optimal choice at the later stages.

Accordingly, $A_u(t)$ would be different for users in each group, for e.g. for a $u$ in LC group, $A_u(t = 1) = 1$, $A_u(t = 2) = 2$ while for EC group, $A_u(t = 1) = 4$, $A_u(t = 2) = 5$ and so on. An example of the linear cascade setting is shown in Fig 4, where a participant receives social signals from its six neighbors—our influence decision $C$ uniformly cascades through the peers of the participant.

At Time Step 13 (start of the second phase), a signaler $u$ selects a sub-optimal provider $C_u$, and over the next five time steps, the remaining peers adopt the same *influence decision* one after another. Note that although we program only selected peers (bots) of a subject to administer $C_u$ over time, the subjects are able to view all of their peers' decisions in their dashboard for the last six time steps. The rest of the non-controlled peers at a time step show random technologies to the participant. Also, note that in all conditions, users can switch back to any choice in the next time step after having selected an option in the current time step. We considered the NM and UM groups as our baseline groups and LC, EC, DC groups as our treatments groups of interest.

For both the hypotheses $H1$ and $H2$, the outcomes of interest are the decisions made by participants in the last six time steps, in the presence of social signals from peers. We explore whether decision-makers will be more likely to choose cyber-defense providers which are not

the optimal choice when they have knowledge about the utilities and when they observe peers opting for choices which are not optimal. We note that people get feedback about their choice on the very next screen, and so choosing a technology during an intermediate time step is more of a data-gathering exploration rather than their final choice. In order to allow for this initial bandwidth for exploration, we keep the first 12 time steps (time steps) the same for all subjects devoid of any interference This helps in overcoming bias related to an individual's own knowledge about the utilities in the second phase of the experiment when they are treated to social signals.

## 4.2 Participants

We recruited a total of 357 participants for this study to play the same cyber-defense provider game. Based on the responses provided by the participants regarding their demographics in the form of a survey response 1 Some participants declined to provide a response regarding their demographics, we have 151 females and 190 males in the study. Most of our participants are between the age group of 26-35 years and full details regarding their age groups have been provided in Tables 1, 2 and 3 in Appendix A in S1 Appendix. Additionally, majority of the participants in our experiment had a Bachelor's degree (See Appendix for more details). Participants were paid \$2 with the opportunity to earn up to \$4.52 since as mentioned before, they received a bonus of \$0.02 for every attack they prevented. Thus, the participants might have motivation to prevent more attacks in order to earn more money. Some of the participants did not complete the full length of the game and so we observed slight discrepancies among the group sample sizes for the statistical tests.

# 5 Analysis

## 5.1 Distribution of attacks prevented

Table 2 shows the distribution of attacks prevented by subjects in each group. We observe that, on average subjects in the EC and LC groups prevent less attacks compared to others. However based on two sampled t-tests, we did not find any statistically significant differences between the groups based on the means of the distributions. Based on a survey analysis, we found that none of the traits like computer anxiety, computer confidence, computer liking, intuition or neuroticism were correlated to the number of attacks prevented in all groups. The details of the survey analysis are presented in Appendix A in S1 Appendix.

## 5.2 Distributions of decisions by individuals

As a first step towards investigating hypothesis $H1$, we analyze the kinds of social signals or the cyber security technologies (which were not the optimal technology) chosen by the peers of each participant and whether they are uniform across all the groups. To simplify nomenclature hereon, we denote the six available technology choices as *decision* $d_i$, $i \in [1, 6]$. In our

**Table 2. Average number of attacks prevented by subjects in each group.** The lower attack numbers suggest participants deviated more from the optimal decision responding to social influence.

| Group | # participants | Average number of attacks prevented (std. deviation) |
|:---:|:---:|:---:|
| NM | 55 | 104.77 (10.99) |
| UM | 71 | 106.44 (9.46) |
| LC | 79 | 103.87 (9.91) |
| DC | 81 | 104.87 (8.39) |
| EC | 71 | 103.50 (8.38) |

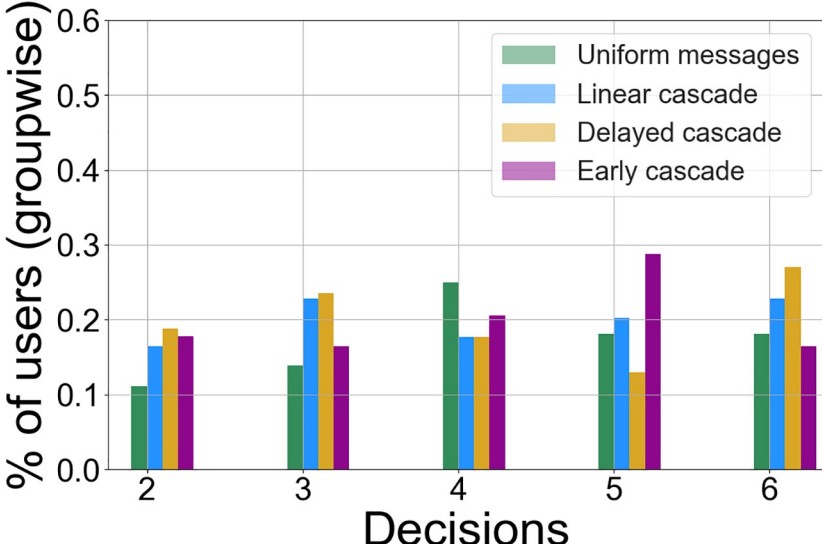

**Fig 5. $P(d_i)$—Proportion of users in each group who were administered technology or decision $d_i$ as the *influence decision*.** Note that only the decisions that are not optimal are sent as prospective influence decisions/social signals in the second phase of the experiment.

experimental setup and for this work throughout, we would refer to $d_1$ as the influence decision—It is the optimal choice preventing seven attacks, while the rest of the technologies prevented five attacks and are termed as sub-optimal choices. As mentioned in Section 4.1, the *influence decision* $C_u \in [d_2, d_6]$ is randomized for each $u$, so we first observe the distribution of the sub-optimal decisions as the choices for $C_u$. For each group, we define $P(d_i) = \frac{|\{u \mid C_u = d_i\}|}{|u|}$ as the proportion of users in the group who were administered technology or decision $d_i$, $i \in [2, 6]$ as the influence decision in the second phase. From Fig 5, we observe that the random selection of $C_u$ introduces some disproportionate values of $P(d_i)$ among the groups. For the UM group, around 25% of users were administered $d_4$ as the influence decision $C_u$ (this proportion $P(d_4)$ for UM is the highest among all other decisions) while 29% of users in the EC group were sent $d_5$ ($P(d_5)$ being the highest for EC) as their infuence decisions $C_u$ and 28% of users in the DC were sent $d_5$ ($P(d_5)$ being the highest for DC). However, we see that for the LC group, $P(d_i)$ was similar for all decisions $d_i$ that could be selected as $C_u$.

Having observed that there was not one pre-programmed peer choice $C_u$ as the strategical obvious sub-optimal choice across all groups, we proceeded with investigating $H1$. In order to detect any implicit occurrence of a selection bias over the participants, we analyzed whether there is any significant difference in the groups with respect to choices made in the first 12 time steps. To accomplish that, we plotted the probability that an individual makes each decision when aggregated over the first 12 time steps. We found that there is clearly no evidence of differences in the mean statistics of the distributions of all choices between the treatments groups (LC, DC, EC) and the control groups (UM, NM) (Refer to Fig 1 in Appendix B in S1 Appendix). Before we conducted pairwise *t*-tests to check for differences between the control and treatment groups for the decision distributions made by the participants in the second phase, we conducted three one-way ANOVA tests considering the sample means of the two control groups and each treatment group, one at a time. We conducted these three tests for each decision from $d_1$ to $d_6$. The null hypothesis for each test constituted the situation where the means of the number of times a decision was chosen by the participants belonging to the

two control groups and one of the treatment groups, are the same for the technology or decision in consideration. We find the following significant results: for the LC group, we find that for $d_4$, the null hypothesis is rejected ($F(2, 203) = 3.7$, $p = .03$). Similarly for the DC group, we reject the null hypothesis for $d_4$ as well ($F(2, 205) = 3.01$, $p = .04$) and for the EC group, we find differences approaching statistical significance for $d_5$ ($F(2, 195) = 2.99$, $p = .05$). We will come back to this case of EC group shortly while discussing the differences. These tests shed light on the differences in adoptions between the control and treatment groups that occurred in the second phase of the experiment in the presence of social signals.

Following this, we conducted two sampled t-tests to measure the differences in the distributions of decisions adopted by users between pairs of control and treatment groups. We conducted two sets of tests for the two phases of the experiments. The null hypothesis constituted the situation where the means of the decision selection distributions of the two groups being tested for, are not different. The results ($p$-values for each treatment group with respect to the control groups) in Tables 4, 5 and 6 in Appendix C in S1 Appendix suggest no significant difference in the distributions among the groups. This rules out any bias among the participants themselves in the absence of externalities. However, in the second phase of the experiment (time steps 13 to 18 aggregated), we find differences in the selection patterns among the decision-makers in their respective groups. We find the following observations from Fig 6(a) and 6(b) for our treatment groups (see tables in Appendix C in S1 Appendix):

1. **LC**: With respect to the NM group, there are no statistically significant differences in the decisions taken by the participants in LC. We carried out a similar statistical test comparing the group pairwise means as done for the first 12 steps. On the other hand, we find that there is a statistically significant difference for the LC group participants ($M = 3$, $SD = 2.4$) in the means of the distributions compared to the UM group participants ($M = 3.8$, $SD = 2.53$) for the optimal decision or $d_1$ ($t(149) = 1.9$, $p = .04$) at $\alpha = 0.05$. The difference shows that a significantly reduced number of participants are tempted to choose the optimal technology provider in the presence of linear cascading signal pattern than when a single signal is sent across all time steps.

2. **DC**: When considering the number of times participants choose $d_4$, we find that the participants in the DC group ($M = 0.58$, $SD = 1.08$), differ from the UM group ($M = 0.30$, $SD = 0.61$) and this difference is statistically significant ($t(155) = -2.02$, $p = .04$). However, with respect to NM group or UM group participants, we observe that the users do not differ in their selections when it comes to choosing the optimal provider. Also it does not differ with respect to the most common choice among the peers for DC group—the Decision 6 shown in Fig 5.

3. **EC**: When we considered the choice of $d_5$, the technology that was administered to the majority of the EC group participants, the users in this group ($M = .79$, $S = 1.08$) differ from the NM group ($M = 0.38$, $SD = 0.66$) in their selection and the difference is significant ($t(128) = -2.7$, $p = .007$). We found similar significant results from our ANOVA tests prior to this analysis and this is a successful case of social influence when considering the macro-adoption process for the group as the users not only steered away from the optimal choice, but they also steered towards the decision of their peers.

This suggests that while the social signal to some extent influences an individual in the LC group to deviate from the optimal choice, it does not always translate to the influence decision that was chosen for the corresponding individual. It rather gears the user towards more exploration. However, an early burst of signals in the EC successfully translates towards social influence wherein on aggregate majority of the users sway towards the influence decision more. As

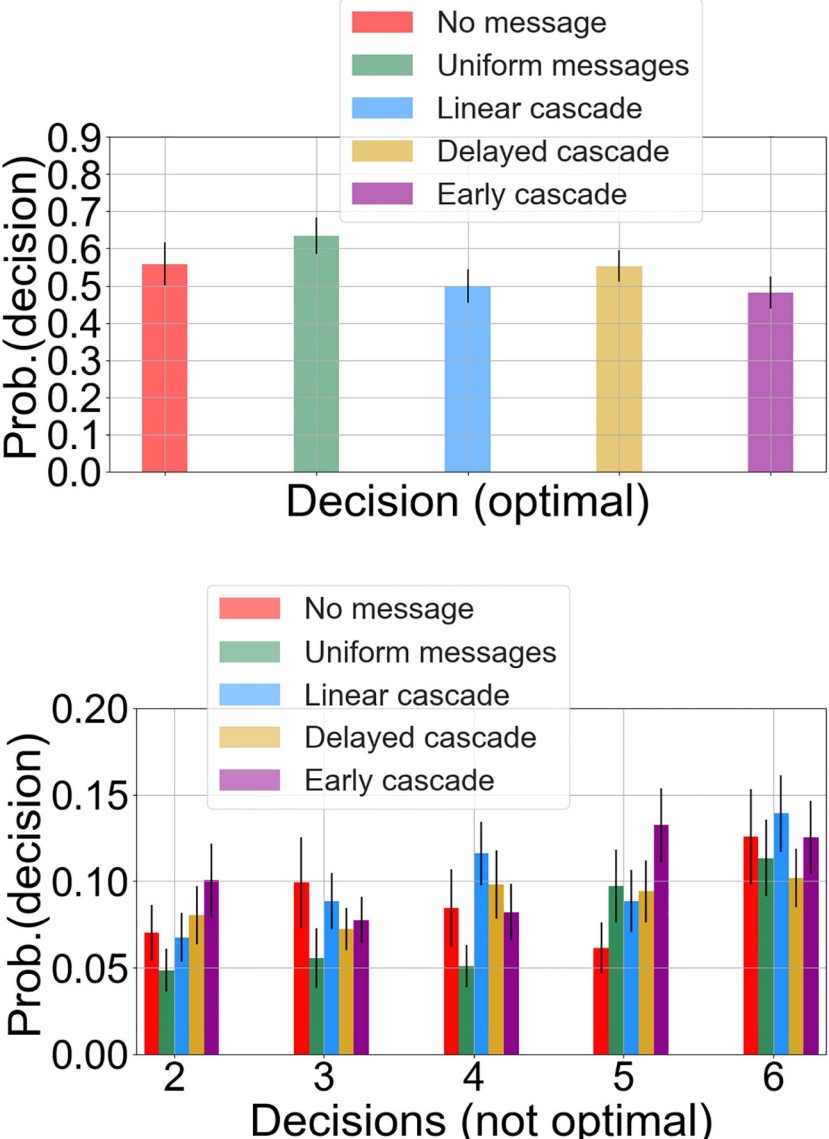

**Fig 6. Probability of decisions made in time steps 13 to 18.** (a) Probability of making the optimal decision. (b) Probability of making the sub-optimal (other 5) decision. The error bars denote standard error over the distributions.

a side experiment to measure the degree of drift away from optimal decision, we also analyze the fraction of users who shift away from $d_1$ (the optimal) at each time step similar to what has been shown in Fig 7. However, we do not find any clear distinctions among the groups in terms of the fraction of subjects who move away from the optimal decision $d_1$ when aggregated over all the six time steps in the second phase. Just the fact that all users eventually move away from the optimal decision when exposed to social signals does not contribute much in distinguishing the PoI.

## 5.3 Degree of influence

This first analysis of H1 does not shed any light on the temporal variations in the decision making process exhibited by users in different groups. It shows some statistically significant

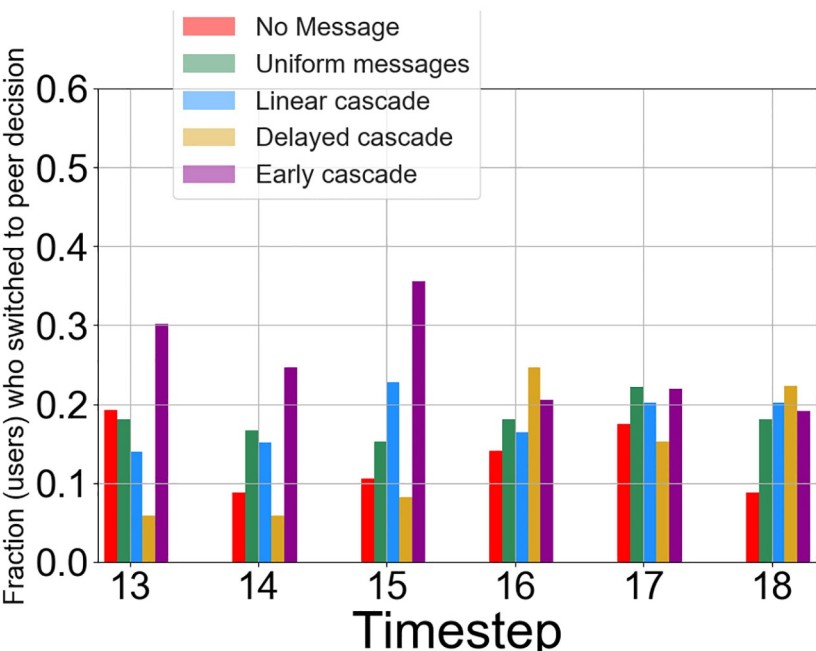

**Fig 7. Fraction of users in each group adopting the *influence* decision chosen by their peers.**

differences in the choices made for specific decisions (or technology providers) over the entire second phase. While it did show that not all cascade patterns successfully influenced users towards deviating from "decision1" or the optimal provider, it brings up the question that is posed for hypothesis *H*2: what constitutes successful influence and does the manner in which the signals are sent determine successful influence?

To this end, we analyze how the proportion of users who switch to the *influence decision* ignoring the optimal choice, evolves over the times steps. We note that for each user *u*, the *influence decision* $C_u$ is randomly selected before the second phase starts. Fig 7 shows the fraction of users in each group adopting the influence decision from time steps 13 to 18, when the experimental participants observe their peers' decisions. From the observations regarding *H*1, we find that the probability of successful influence for participants in EC has the strongest effects on decision-making in the early stages of the second phase. Participants in EC are most likely to deviate from selecting the optimal provider as shown in Fig 6—for the first three time steps in the second phase, participants in the EC group exhibit the maximum adoption compared to other groups denoted by a higher fraction of adopted participants. Additionally, for the EC group, Time Step 15 had the maximum retention where 35% of users adopted their peer behavior, and when participants were exposed to all six of their peers selecting the same technology $C_u$. This may be due to new users adopting the *influence decision* or due to cumulative build-up from previous time steps who do not switch back. We will explore this in the following sections.

The participants in the DC group exhibit successful response towards the sudden increase in signals at Time Step 16 which is shown by a 65% increase in user adoption of the *influence decision* compared to the previous step. These results become close to the 25% of users making influence decision selection at Time Step 16 and which is also the maximum among all groups surpassing the early cascade adoption ratio. However, there is no substantial increase in the adoption fraction for the users in the linear cascade group—we do note that the adoption

peaks at Time Step 15 for the linear cascade users before it drops again. These observations from Figs 6 and 7 suggest that while an early burst of social signals successfully persuades users in EC to adopt the influence decision, causing an aggregated overall maximum selection of the influence decision, a sudden impulse in the quantity of social influence also successfully steers users towards the influence decision in the later stages.

## 5.4 Measuring the effect of quantity of signals

In this section, we investigate whether the quantity of signals alone stand out as the sole factor of influence. Before going into the analysis, we define a few notations: we denote a subject in this study as $u$ where $u$ can belong to any group. We denote the decision taken by a subject $u$ at time $t$, $t \in [1, 18]$ by $D_u(t)$. We define $T_{treat}$ as the sequence of time points during which the subjects receive signals from their peers i.e. $T_{treat} = [13, 18]$. Following this, we denote the time step $t$, $t \in T_{treat}$ when an individual $u$ *first* switches to $C_u$ as $t_u^f$. For each signal quantity $s$, we measure the proportion of individuals (in each group) who made their **first** switch to their influence decision only after they were exposed to $s$ signals. Formally, for any

$$t \in T_{treat}, R(s) = \frac{|\{u \mid D_u(t)=C_u \bigwedge A_u(t)=s \bigwedge t=t_u^f\}|}{|\{u\}|}$$ ($R(1)$ in the LC group denotes the proportion of individuals in LC group who made their first switch to their peer influence decision after being exposed to just one signal. Similarly, $R(6)$ in the EC group denotes the proportion of individuals in EC group who made their first switch to their peer influence decision only after being exposed to six signals, so this can happen in any of the last four time steps in EC). The denominator in the formula here denotes the number of individuals in the group.

We bin the values from $R(s)$ based on $s$ and take the mean for each group, since for some groups, there can be multiple time steps with the same number of exposures or signals. From Fig 8(a), we observe that for the EC group, $\sim 30\%$ of users under the influence of four signalers, for DC, $\sim 18\%$ of users at four signalers and for LC, $\sim 15\%$ of users at three signalers (all these being the maximum ratio) made their **first** switch to the influence decisions in the second phase of the experiment. However, on close observation, we find that the number of

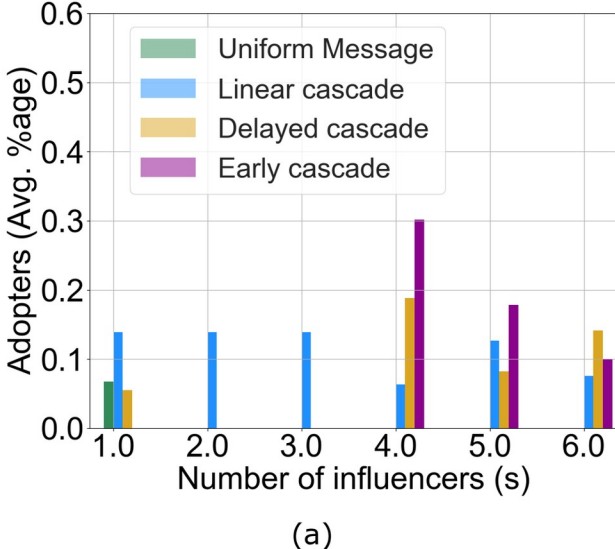 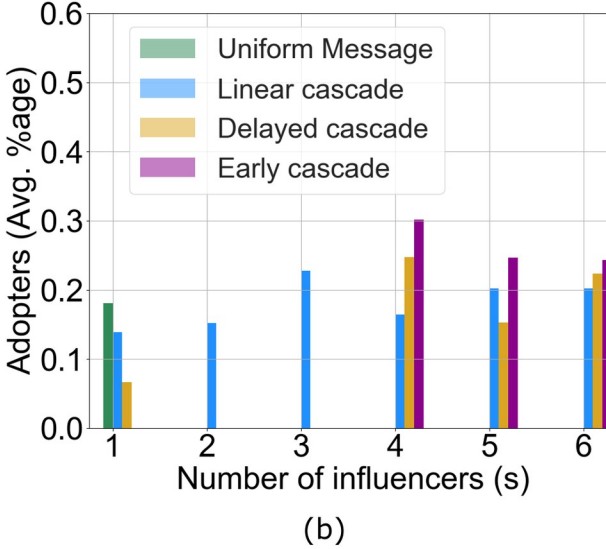

**Fig 8. Plots of adoption under the influence of *s* exposures.** (a) For each signal quantity *s*, the proportion of individuals who made their **first** switch to their influence decision after being exposed to *s* signals, (b) The proportion of individuals who adopt their influence decision after being exposed to signals from *s* influence signals.

exposures alone does not explain the adoption behavior. When we compare the EC participants with those in the DC group, we find that with four exposures in $T_{treat}$ (at Time Step 13 for EC and at Time Step 16 for DC), the proportion of adopters in EC making their first influence decision switch (30% of users) is higher than the proportion in DC (18% of users), although the same four quantity of signals are delivered at different time steps for the two groups. However, while there is a constant decrease in the number of adopters making first switches in the EC group going from four to six exposures denoted by the corresponding mean $R(s)$, we see that the same quantity for the DC group does not decrease the same way for four to five to six exposures. This suggests that the sudden stimulus from the delayed exposure somehow succeeds in influencing more users to make their first switch to the influence decision compared to the EC group (note that all the users under the quantity $R(s)$ for each $s$ are unique since we measure their first switch). On the other hand, for the linear cascade group we do not find one quantity that is most effective in the influence. In the LC setup, there is no one exposure that impacts the adoption behavior the most in terms of successful influence. In addition, we measure the cumulative adoption ratio for each group defined as: for any $t \in T_{treat}$,

$Z(s) = \frac{|\{u \mid D_u(t)=C_u \bigwedge A_u(t)=s\}|}{|\{u\}|}$. In simple terms, it measures the number of individuals in a group who adopt the influence decision under $s$ exposures irrespective of whether it is the first switch. This is demonstrated in Fig 8(b). When we combine the results obtained in Fig 8(a) with Fig 8(b), we find an interesting observation for the LC group. The linear cascade pattern is able to retain most of the users even after first switch at Time Step 13 as the cumulative ratio increases up to three exposures (which occurs at Time Step 15). This suggests that the LC pattern is effective in terms of retention in the early stages of the cascade.

## 5.5 Quantitative analysis using growth modeling

To understand the patterns of change over time by incorporating the heterogeneity among individuals in each group and among the groups, we resorted to the widely used statistical tool of growth modeling through random coefficient models [39]. Briefly, the technique of growth modeling allowed us to test the longitudinal effects of the peer signals on the users and test any source of heterogeneity in decision making among individuals that lead to the observed outcomes. One of the advantages of growth modeling comes from treating time as a predictor of influence outcome in the absence and presence of the peer signals. The error-covariance matrix from these models informed us about the variations among the individuals in the presence and absence of peer signals over the phases of the experiment.

We utilized four regression models where the outcome of interest denotes whether an individual selected the *influence decision* $C_u(t)$ at time $t$ in the second phase of the experiment. Let the linear predictor, $\boldsymbol{\eta}$ be the combination of the fixed and random effects excluding the residuals. Considering a linear mixed effects model LMEM $\mathbf{y} = \mathbf{X}\beta + \mathbf{U}\boldsymbol{\chi} + \epsilon$, where $y \in \mathbb{R}^k$, (and $k = n * [t]$) denotes the outcome response of individuals over all time points $t \in [1, [t]] - k$ thus denotes the number of instances in the model. In this work, we have $[t] = [T_{treat}] = 6$. $\mathbf{X} \in \mathbb{R}^{k \times f}$ denotes the design matrix of endogenous variables, $f$ being the number of predictors, $\mathbf{U} \in \mathbb{R}^{k \times w}$ denotes the matrix with random effects (the random complement to the fixed $\mathbf{X}$), $w$ being the number of predictors with random effects, $\epsilon$ denotes the residuals, and $\beta$ and $\boldsymbol{\chi}$ are the parameters of interest corresponding to the fixed effects and the random effects in the model. Since the outcome in our work is binary, we used Generalized Linear Mixed Effects models (GLMEM) [40] in this work. In a GLMEM model, we used the same equation as LMEM but with a linear predictor $\boldsymbol{\eta}$ such that $\boldsymbol{\eta} = \mathbf{X}\beta + \mathbf{U}\boldsymbol{\chi} + \epsilon$ where $g(\mathbf{E}[\mathbf{y}]) = \eta$, $g(.)$ being the link function. To accommodate for the binary outcomes, we used the logit link function.

To quantify the effects using growth modeling, for each individual, we used the following regression models for modeling the growth functions:

$$\mathbf{M0 - FE} : \eta_i = [\beta_{00} + \beta_{10}(Time_i)] + \epsilon_i \tag{1}$$

$$\mathbf{M1 - RE} : \eta_{ij} = [\beta_{00} + \beta_{10}(Time_{ij}] + \chi_{0j} + \epsilon_{ij} \tag{2}$$

$$\mathbf{M2 - RE} : \eta_{ij} = [\beta_{00} + \beta_{10}(Time_{ij}] + [\chi_{0j} + \chi_{1j}(Time_{ij})] + \epsilon_{ij} \tag{3}$$

$$\mathbf{M3 - RE} : \eta_{ij} = [\beta_{00} + \beta_{10}(Time_i) + \beta_{20}(Signals_i)] + [\chi_{0j} + \chi_{1j}(Time_i)] + \epsilon_{ij} \tag{4}$$

where the indices $i$ denote the observation instance number (or the row number in a table of data) and $j$ denotes the individual in the group of all users. So, **M0-FE** represents the fixed effects model where the only independent variable is the time. Intuitively, this model tests how individual responses to peer signals evolve over time, when time itself is the only factor in consideration. From the results in Table 3, we find that among the three treatment groups, the probability of outcome is most positively correlated with time for the Delayed Cascade group. This simple regression model ignores the fact that the observations are nested within individuals and accordingly, the next step in growth modeling is to add a component of random intercept to the model—this is denoted by **M1-RE**. Note that in this model, the random intercept $\chi_{0j}$ is specific to each individual $j$. On comparing the parameter estimates, we find that the time coefficient $\beta_{10}$ remains similar for both models even after incorporating between-person differences for all the groups. This shows that time is significantly correlated to the outcome in the absence of the knowledge of peer signal treatment.

Next, we added the "Time" variable to the random components, so that time could randomly vary among the users. This is denoted by the model **M2-RE** given by Eq 3. The correlation between the slope and intercept for this model for the LC, DC and EC groups are respectively 0.7, 0.7 and 0.4 respectively. The positive correlation indicates that individuals who have a high propensity to move towards the *influence decision* in the beginning tend to have strong slopes—this weakly suggests that individuals with some degree of uncertainty towards the optimal decision at the beginning tend to be more susceptible to influence over time. Next, we explored the idea that "Signals" have a role to play on the outcome of the individuals. We added to M2-RE, the fixed effects from Signals shown in **M3-FE**. From the results in Table 3, we found that for all the three groups, the number of signals is positively correlated with the outcome while the time factor is negatively correlated for the groups. Among all the groups we found that for the DC group, the Signals factor has the highest coefficient suggesting the strong correlation between the treatment with signals and the outcome. These results suggest a positive correlation of the treatment of signals on the outcome when time has an important role to play. The interaction between time and signals are important here given that the correlation of time changes once the effect of signals is considered.

**Table 3. Results of Fixed Effects (FE) and Random Effects (RE) modeling.** "Inter". denotes intercept in the regression models. Values in brackets denote standard errors.

| | Linear Cascade | | | Delayed Cascade | | | Early Cascade | | |
|---|---|---|---|---|---|---|---|---|---|
| | Time | Signals | Inter. | Time | Signals | Inter. | Time | Signals | Inter. |
| **M0-FE** | 0.08 (0.07) | | -1.79 (0.28) | 0.32 (0.08) | | -3.08 (0.36) | -0.12 (0.06) | | -0.67 (0.24) |
| **M1-RE** | 0.07 (0.03) | | -1.91 (0.12) | 0.32 (0.03) | | -3.23 (0.13) | -0.13 (0.03) | | -0.69 (0.11) |
| **M2-RE** | 0.05 (.03) | | -1.83 (.12) | 0.29 (.03) | | -3.15 (.13) | -0.15 (.03) | | -0.63 (.11) |
| **M3-RE** | -0.02 (.27) | 0.29 (.23) | -1.87 (.96) | -0.37 (.029) | 0.44 (.074) | -0.23 (.025) | -0.23 (0.12) | 0.29 (.25) | -0.28 (.42) |

## 5.6 Interplay between influence and susceptibility

We ended this study with a retrospective analysis to understand the dynamics of adoption under a slightly relaxed setting. In real-world networks, not all individuals are susceptible to social change emanating from their neighborhood—some people have stronger beliefs than others [41]. In an attempt to quantify the effect of the influence on subjects in a more constrained setting, we considered only those users $u$ who have been influenced to adopt the *influence decision* $C_u$ at least once between time steps 13 to 18 in the second phase. We measure at every time step, the ratio of individuals who adopted the influence decision at that time step to the number of individuals in the group they belong to, i.e., who switched to their influence decision *at least once* within their lifecycle ($T_{treat}$). Note that this is different from previous measures in two ways: first we retrospectively filter out users who never adopted their influence decision (in the real world these are users who would not be susceptible to influence or are immune as such). Second, we analyze this ratio at the end of their exploration phase, in Time Step 18, when everybody has supposedly settled down. We define a symbol $N_u$ as the number of time steps for which a user $u$ adopts $C_u$ in $T_{treat}$ (this is measured retrospectively aggregating all time steps beforehand). Formally it is defined as: $Success\ ratio(t) = \frac{|\{u\ |\ D_u(t)=C_u\}|}{|\{u\ |\ N_u \geq 1\}|}$. The denominator denotes the number of individuals who have adopted the influence decision at least once from Time Steps 13 to 18. The comparison shown in Fig 9 among the four groups (the No message group does not have any influence decision) demonstrates that while the EC group adopts the influence decision more quickly than other groups, the stimulus in signals quantity at Time Step 16 in the DC group affected the participants. This is confirmed when the effects of DC strongly outstrip those observed from EC in the last time step where both groups receive six signals. At the end of the game, at Time Step 18, we find that the highest number of such susceptible individuals come from the DC group—these results reinforce some of the conclusions we had from Section 5.3 and Fig 7 regarding the late retention capabilities of the DC group strategy—however, this measure makes the differences clearer when we consider individuals who are more prone to social influence.

## 6 Data-driven computational models

In this second study, we extend our work on understanding the behavioral aspect of responses to social influence in sub-optimal choice diffusion settings to real world scenarios where the distinction between the utilities among choices may not be outright evident. Additionally, as mentioned in the introduction, it is generally not easy to observe these patterns of influence at scale and also in networked settings in the real world which makes it difficult to study the effects it might have on behavior diffusion. The opaque nature of the effect of exposures in the real world responsible for influence makes it more difficult to analyze the characteristics of these PoI. The fact that the data about who-exposed-whom in real world information cascades is rarely available makes studying peer influence more challenging [16]. To this end, we tried to bridge the gap between experimental hypothesis and real-world scenarios by modeling the behavior of agents or users when subject to peer influences and by capturing the sequential nature and bursts in influences towards diffusion. We took the case of rumor diffusion when the piece of information that propagates as a rumor turned out to be false. In such cases, the action of resharing by individuals is sub-optimal from the perspective of information sharing. In such situations, social influence plays an important role in persuading individuals with benign intent towards resharing when in fact these individuals might otherwise be reluctant to participate. We observe the trade-off between individual decisions and the influence decision

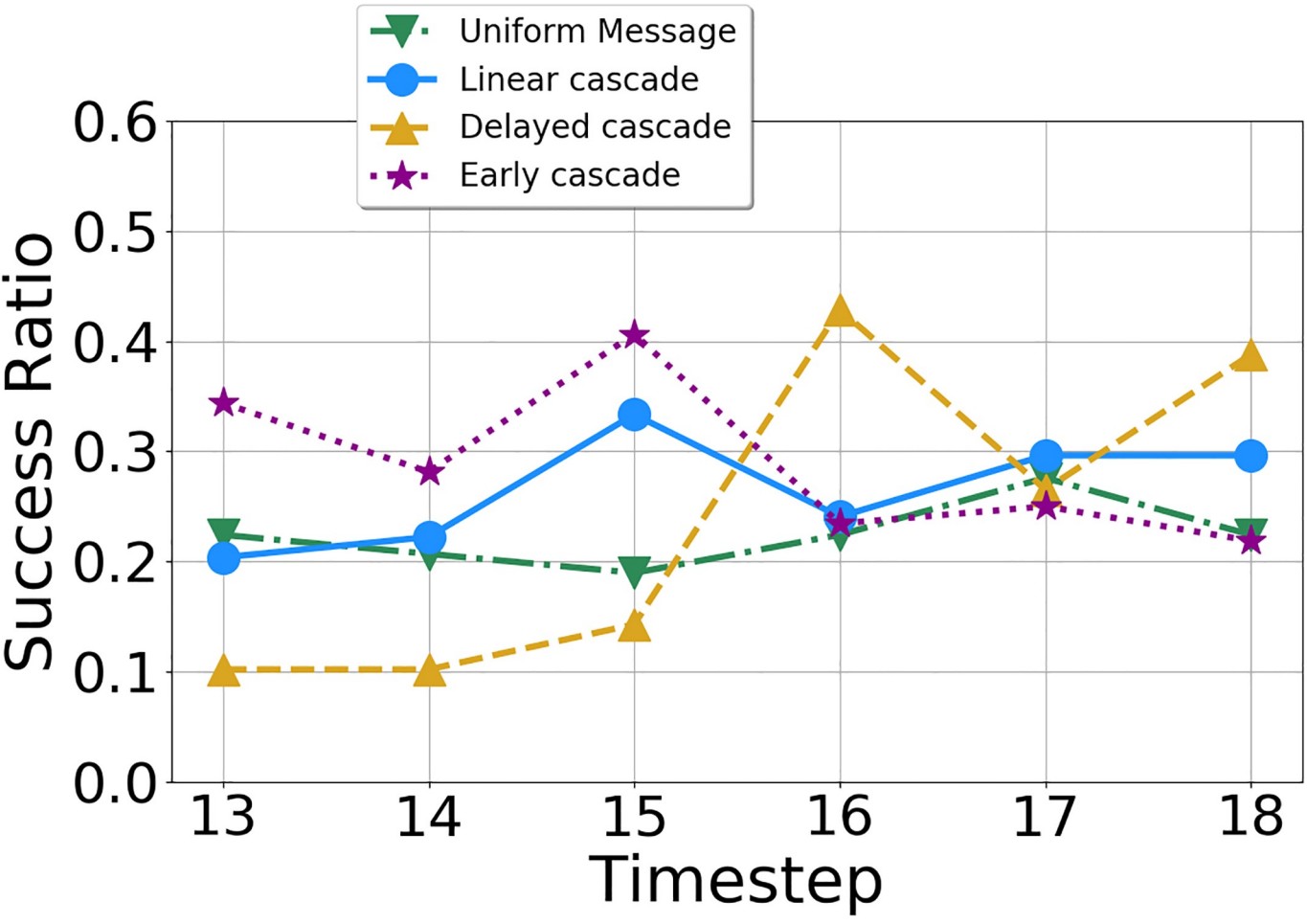

**Fig 9. Success ratio.**

through the cascading effect from peers in such environments where resharing a message would be the wrong choice.

While the growth models in Section 5.5 provided us with evidence about the importance of signals across time for the three treatment groups, the analysis from Section 5.4 showed that the compounding effect of influence can lead to different probability outcomes. Specifically, we found that the number of signals as the proxy for influence can have different effects when it comes to users adopting the influence decision. We observed the values of $R(s)$ being different among the LC and the EC groups for the same $s$ signals despite their being administered at the same steps of the game. This non-markovian nature of influence calls for developing models that not only take the effect of the magnitude of peer signals into account but also its effect relative to what the user has been exposed to across the time steps thus far. However, observing such influence patterns is not always trivial when it comes to mapping and filtering these chains of patterns in real world cascades. The challenge was exacerbated when we tried to weigh the users' private information against the factor of influence. We did not observe these private cognitive factors in real world data which makes it more difficult to understand the real world implications of our experimental conclusions.

To this end, we defined our models of influence taking into account the impact of sequential exposures and performing simulations of the spread of adoption using an influence based multi-agent model on real world data. This also helped us measure the extent to which the observations from our controlled setup can be replicated in real world cases through simulations. Our agent-based model (ABM) differs from the traditional models of diffusion in that (1) agents as influencers are homogeneous unlike in traditional models, where each influencer makes its own contribution towards the group influence function and therefore in our case, pairwise influences between users and their peers were similar for all peers and (2) behavior diffusion in the simulation was directed towards sub-optimal decision making where the choice of following peers or the *influence decision* may not be the optimal decision.

## 6.1 Simulation setup for ABM

There are two ways in which we can model agent behavior to simulate behavior diffusion—the *single agent* behavioral model which predicts individual behavior when individuals are observed in isolation and the *multi agent* behavioral model that extends the *single agent* behavioral model to a population level by executing the simulation in a multi-agent environment. In such scenarios, since the agents influence each other through their own actions over a period of time, the behavior of individuals can no longer be measured without taking the environmental factors into account. Fig 10(a) shows the agent-based model for a single agent where an agent makes a decision based on its utilities at every time step. Once the utilities have been determined, the agent picks a decision corresponding to the maximum utility. Once the agent has acted, the environment is updated and external factors that might also impact the agent's

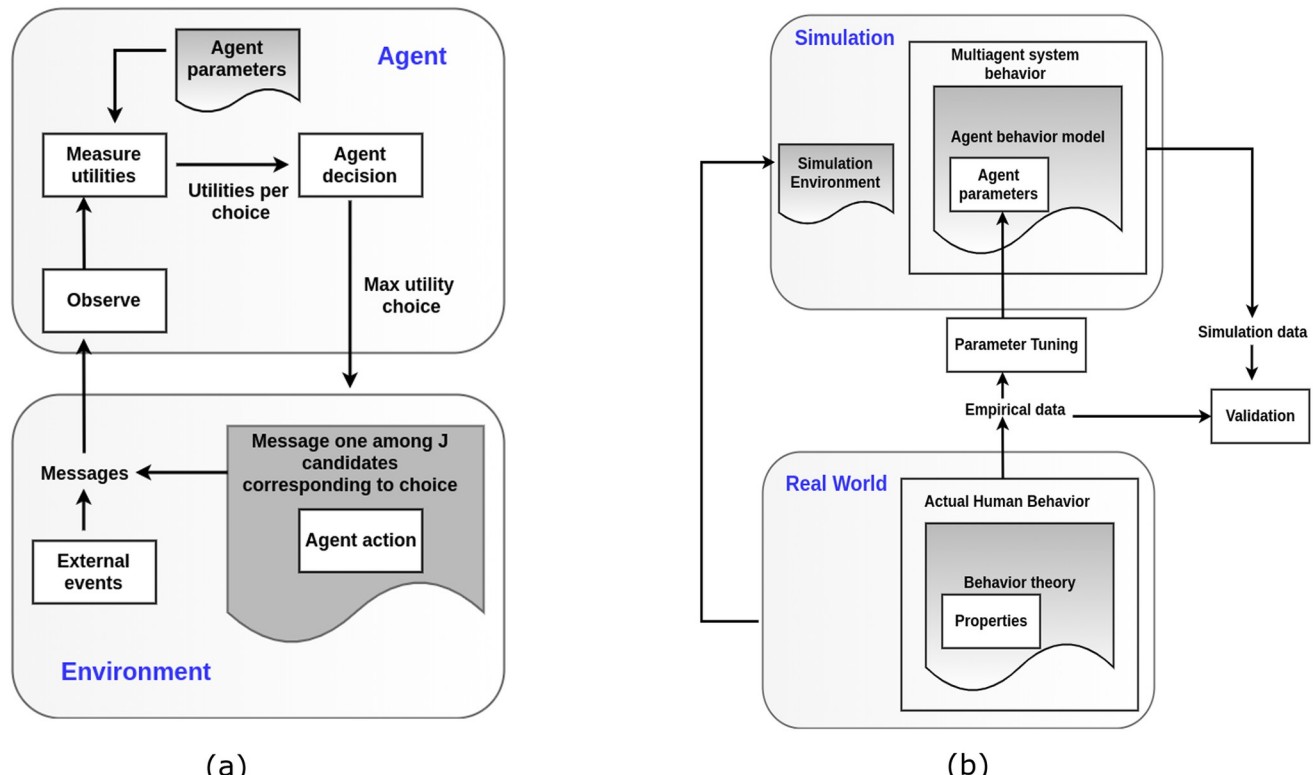

(a)                                                                  (b)

**Fig 10. Agent based models for measuring social influence with multiple choices.** (a) Single agent behavior model, (b) Multi-agent behavior simulation.

decision in the next step is accounted for. Prior to the next time step, the utilities of the choices are updated based on the previous decision. However in this lifecycle of the decision making process, the agent is observed in isolation.

In our work, we used this single-agent behavior model as a generative element to simulate agent behavior but we observed the behavior at a population level especially when the agents influenced each other by making actions and by virtue of being in a networked environment. In order to simulate and evaluate the effect of social influence through multiple exposures in the real world, we chose a specific real world case study to implement the environment. As shown in Fig 10(b) which represents the multi-agent behavioral model lifecycle, agent behavior is simulated using the single agent behavior model in Fig 10(a) with specific input parameters for the agent which would be learned from the empirical data. We ran the ABM and evaluated the results from the ABM using different training and test splits but from the same distribution of the empirical data. The agent specific parameters were learned prior to the start of the simulation iterations since we mapped the agents to real world users. Although we did not use feedback from the evaluation obtained from one run of the simulation from the real world to optimize the simulation environment for future runs, this step can be additionally performed to obtain monotonically increasing evaluation performance. In our environment, each agent has a choice to reshare a piece of information in the situation where resharing is not the optimal choice. The following sections describe the agents and their choices in details.

## 6.2 Agents

In our multi-agent simulation environment, the agents and the users in the social network of the real world are one-one mappings and so by default the agents are embodied in a networked environment where they can share and be exposed to others' messages. We modeled a social network as a directed graph $N = (V, E)$, where a node $v \in V$ represents and individual and the edge $(u, v) \in E$ exists if $u$ follows $v$, in our case $v$ influences $u$'s decision. So in the context of this social network, our agents are the nodes in these networks. Each agent can play both the role of a neighbor influencing a user or a susceptible user who is being influenced. For each agent $u$, as mentioned before, we considered the set of agent's peers to be homogeneous with respect to the influence they exerted on $u$. In the context of the ABM, we separately modeled the probability that each agent was being influenced based on factors that are going to be discussed in the next section.

## 6.3 Modeling agent behavior towards sub-optimal decision making

Instead of considering multiple choices for the agent decision stages, we considered a binary choice model where at each time step, each agent had to make a selection among two choices which are reciprocal to each other, and the selection is based on the choice that comes with the higher probability of activation. In the real world case study used for the simulation, the action of resharing a piece of information is a sub-optimal choice and so the peers of an agent who reshared the same piece of information exerted influence towards sub-optimal decision making.

Also, note that this is a simpler version of the online controlled experimental setup where the user had five choices. One of them was optimal and our setup could be extended to include multiple choices for a relevant real world case study. Following this, the agent could only be in two states based on the choice they made, that is they have either reshared a piece of information identifying the cascade or they have not. Additionally, as in most real world adoption scenarios with a binary choice model, the user cannot transition back to the state it arrived from. Before delving into the technical details of the components for the agent utilities, agent states

and the exposure effect based ABM, described in details below is how we quantified the individual decision factor and the peer influence through exposures.

**6.3.1 Probability of activation.** The agent in our model starts with being agnostic about whether the choice of resharing a message or the activation choice is the optimal choice, however it is in the interest of a rational actor to make a choice that is optimal in the real world, yet conform to the general choices made by the population. So in the absence of any external factors, our model posits that the probability that agent $u$ makes a sub-optimal decision is given by:

$$p_u(t) = \frac{1}{1 + exp(-(\zeta_u(t) - \mu_u(t)))} \qquad (5)$$

where $\mu_u(t)$ denotes the negative utility from the agent $u$'s own decision to select the optimal choice at time $t$ using its acquired knowledge of the real world event (this quantity represents the utilities from the knowledge acquired by users in the first phase of the controlled experiment) i.e. it represents the cost from disagreeing with the dominant influence decision which is not optimal, and $\zeta_u(t)$ denotes the utility that comes from selecting the peer choice $C_u$. An important point to note here is that while $\mu_u(t)$ denotes the loss from selecting the optimal decision, the utility $\zeta_u(t)$ denotes the utility from adopting the *influence decision* and we incorporated that in the function definition described later. So $p_u(t)$ here denotes the probability of selecting the sub-optimal decision which is key to the way we handled the simulation later.

Note that unlike other propagation models [42–44], we did not consider external effects for the propagation like infectiousness parameter of the cascade, time of the day as those factors can be added to our ABM model as well the baselines we use to measure the effect of exposures as a proxy for influence. The goal of this ABM model based simulation of spread is two fold: (1) we measured the extent to which the spreading on real world networks based on the exposure effect from our model differs from that of the Bass model and which was the basis of the controlled experiment in [17] which considers only the number of active individuals (using the network structure to diffuse signals) at a time as a measure of influence, and (2) how close the results obtained from the spreading simulation are to real world diffusion. We repeat that any other external factors that influence adoption can be added to our model to improve the fit to real world data—however we tested exposure rates for social influence as opposed to just active neighbors, exclusively through this ABM. Next, several models are described that define these two probability measures in Eq 5.

**6.3.2 Individual decisions.** Most of the diffusion models that measure the impact of behavior are somewhat mechanical and not strategic, meaning that the probability that an agent adopts a specific behavior is proportional to the infection rate of her/his peers. However, the controlled experiments showed that social influence based exclusively on the quantity of peer signals at any time does not always determine the outcome and that individual choices of rational agents can determine the diffusion process quite significantly. Additionally, each agent wanted to maximize her/his utility through intentionally selecting behaviors. Intuitively, considering the stochastic and non-stationary nature of human decision, it is essential to accommodate uncertainty when users infer utility from interactions. We achieved this individual decision component through the following latent variable. Let the utilities associated with the choice of making the optimal decision be given by a latent unobserved variable $x_u(t)$ that determines the individual utility that drives user $u$'s decision making at time $t$. Since in most real world studies, there is no straightforward way to determine the individual intentions behind resharing a message, we used this variable to capture it. In the context of sub-optimal

decision making, this utility from an agent's standpoint towards optimal choices now counts as a loss towards the net utility for making sub-optimal decisions.

We do note that as mentioned in several existing studies [45], there can be several other factors like the infectiousness of the current event and other intrinsic factors of the user that contribute to the utility. We repeat that all these factors can be incorporated to make the model more realistic. The utility that a user gets from making the optimal decision is then

$$\mu(t) = x_u(t) + \epsilon \tag{6}$$

Here $\epsilon$ is an i.i.d. random variable drawn from some generating distribution that accounts for the uncertainty in the behavior of individuals beyond their own utility for a decision. We started the simulation after already observing the initial set of reshares for a cascade. Following this, we consider that the utility a user gains from selecting the optimal choice as per its own knowledge is constant over time after the initial stage.

**6.3.3 Utility from influence decision.** Social influence phenomenon arising out of individual interactions is measured through pairwise influences that result in a complex contagion. The basic assumption is that the probability of an agent being activated is dependent on the heterogeneous pairwise probabilities between the agent and its peers. In the simplest case for a specific individual, when we only measured the number of peers who had already adopted a particular message as a measure of social influence [46], the probability $p'_u(t)$ that $u$ is activated at time $t$ is given by:

$$p'_u(t) = \frac{1}{1 + exp(-[\eta_u \ A_u(t) + \beta_u])} \tag{7}$$

where $\eta_u, \beta_u$ are coefficients to be estimated. Equivalently,

$$\zeta_u(t) = ln\left(\frac{p'_u(t)}{1 - p'_u(t)}\right) = \eta_u \ A_u(t) + \beta_u \tag{8}$$

where coefficient $\eta_u$ measures the social influence or social correlation effect for $u$. Intuitively, the right hand side of Eq 8 denotes the utility of the agent $u$ obtained from adopting the *influence decision* in situations where resharing is not the optimal choice.

One of the key observations from the controlled experiments shown in Fig 8(a) and 8(b) is that the slow compounding effect on behavior outcome from linear influence cascades may not be the best in terms of the desired outcome at all time steps. We observe that the sudden spike in signals at Time Step 4 for the delayed cascade participants (when both LC and DC participants had four peer exposures) allows for more users in DC to respond to social influence compared to the LC group. We introduced a scaling factor for the quantity of peer signals that capture this spiking effect. To this end, instead of using the number of exposures directly as the peer effect, we substituted $A_u(t)$ with the following:

$$A_u(t) = A_u(t).e^{\sigma(A_u(t) - A_u(t-1) - 1)} \tag{9}$$

The intuition behind the augmented exposures is that the sudden spike makes an amplifying effect on the social influence measure and so should be accounted for. Here $\sigma$ is the parameter that controls the amplifying exponential curve. Note that for the linear cascade pattern, where there is a single increase in the peer exposure at $t$ with respect to $t-1$ at all times, the amplification is null. The scaling parameter $\sigma$ is held constant for all agents.

**6.3.4 Models of decision making.** Using the two components above, we arrive at two models of activation based on Eq 5 and we used those 2 models to run our simulation procedure:

1. **Base model (BM)**: We used Eqs 6 and 7 to arrive at the following probability of activation

$$p_u(t) = \frac{1}{1 + exp(-[(\eta_u\ A_u(t) + \beta_u) - (x_u(t) + \epsilon)])} \tag{10}$$

2. **Augmented Exposure model (AEM)**: We use Eqs 6 and 9 to arrive at the following probability of activation with the augmented peer exposures

$$p_u(t) = \frac{1}{1 + exp(-[(\eta_u\ (A_u(t).e^{\sigma(A_u(t) - A_u(t-1)-1)}) + \beta_u) - (x_u(t) + \epsilon)])} \tag{11}$$

The two probabilities shown above represent a situation when the agent decides to reshare the message after weighing the utilities from the two components. Since we adopted a binary choice model, we did not explicitly model the utilities of the other choice which is to not reshare the message. The probability of an agent not sharing the message is then just $1 - p_u(t)$.

## 6.4 Learning model parameters

With the models of activation stated as above, we set the parameters of the model in the simulation procedure. We specifically worked with information cascades representative of real world diffusion [47] while discussing the dataset. Since we considered rumor diffusion as the real world study and we calibrated the parameters of the models to this dataset by splitting it into training and evaluation sets as is prevalent in machine learning setups. Specifically, we started the behavior diffusion simulation of the agents after observing part of the diffusion cascades till $T_{thresh}$. That is, we first observed the cascades from their beginning to a specific time span $T_{thresh}$ for learning and left the rest of each cascade after $T_{thresh}$ to be used for evaluation of the ABM. This helps us in two ways: first, it allowed us to perform simulation with the assumption that the agents had the time to form some opinion of their own using the exploration strategy as setup in the controlled experiments. Second, it allowed us to learn the parameters specific to each agent in a data-driven way prior to start of simulation and allows us to perform evaluation of the ABM based diffusion process after $T_{thresh}$.

We treated the latent utility factor $x_u(t)$ as a parameter of interest and considered that the agent's individual decision utility was fixed. We considered $x_u(t) = x_u$ for all time steps $t \in [1, T]$. Specifically, the parameters of interest specific to agents are $\theta_u = \{x_u, \eta_u, \beta_u\}$ and the parameter $\sigma$ which we set to a constant during our evaluation. Since it is not easy to map the controlled experimental environment to situations in observational studies, we did not consider agent histories, so instead of learning individual agent parameters $\{x_u, \eta_u, \beta_u\}$, for each $u$, we divided all the agents into $L$ latent groups. This also captured the notion that agents in a connected network belong to a latent block structure or specifically stochastic block models [48]. In a stochastic block model, each agent is assigned to a block and the pattern of influence between different agents depends only on their block assignment. Following this, the overall probability that $u$ belongs to class $l \in [1, L]$ is given by $p_l = P[l_u = l]$ with $\sum_{l=1}^{L} p_l = 1$. So all the individual agent specific parameters $\{x_u, \eta_u, \beta_u\}$ are now replaced by $\{x_l, \eta_l, \beta_l\}$. Let $\theta_l = \{x_l, \eta_l, \beta_l\}$ be the parameters specific to the latent class $l$. Denoting $z_u(t) = 1$ if the agent reshares the message (sub-optimal decision) and 0 otherwise, $Z_u = \{z_u(t)\}, \forall t \in [1, T]$, the likelihood

contribution of $u$ belonging to latent class $l$ is then given by:

$$f(Z_u; \theta_l) = \prod_t p_u(t)^{z_u(t)} (1 - p_u(t))^{(1 - z_u(t))} \tag{12}$$

Denoting the set of parameters $\Theta = [\theta_1, \ldots, \theta_L]$, $\mathbf{P} = [p_1, \ldots, p_L]$ and $A_u = \{A_u(t)\}$, $\forall t \in [1, T]$, the likelihood of the model is then given by:

$$\mathcal{L}(\Theta, \mathbf{P}|D, A) = \prod_u \left( \sum_l p_l \, f(Z_u; \theta_l) \right) \tag{13}$$

So our log-likelihood is:

$$l(\Theta, \mathbf{P}) = \sum_u \log \left( \sum_l p_l \, f(z_u(t); \theta_l) \right) \tag{14}$$

We attempted to compute the posterior distribution of the parameters given the observations:

$$P(\Theta_l, p_l|A_u, Z_u) = \frac{p_l f(z_u(t); \theta_l)}{\sum_l p_l \, f(z_u(t); \theta_l)} \tag{15}$$

Denoting $W_u = \{A_u, Z_u\}$ for all agents $u$, we first attempted to compute the posterior distribution of $p_{l,u} = P(l_u = l|W_u)$, given the observations. And formally it is given by:

$$P(l_u = l|W_u) = k_{u,l} = \frac{P(W_u|l_u = l)P(l_u = l)}{P(W_u)} = \frac{\pi_l \, f(Z_u; \theta_l)}{\sum_l \pi_l \, f(Z_u; \theta_l)} \tag{16}$$

The lower bound log-likelihood following Eq 14 takes the form

$$\begin{aligned} ll \quad &= \sum_u \log \mathbb{E}_{l \sim k_{u,l}} \left[ \frac{\pi_l f(Z_u; \theta_l)}{k_{u,l}} \right] \\ &\geq \sum_u \sum_l k_{u,l} \log \frac{\pi_l f(Z_u; \theta_l)}{k_u^l} \end{aligned} \tag{17}$$

Taking the derivative of $ll$ with respect to $x_u$ and keeping other parameters fixed, we get

$$\nabla_{x_l} ll = \nabla_{x_l} \sum_u \sum_l k_{u,l} \log \frac{\pi_l f(Z_u; \theta_l)}{k_u^l}$$

$$= \nabla_{x_l} \sum_u \sum_l k_{u,l} \log \frac{\pi_l \left[ \prod_t p_u(t)^{z_u(t)} (1 - p_u(t))^{(1-z_u(t))} \right]}{k_u^l}$$

$$= \nabla_{x_l} \sum_u \sum_l \left[ k_{u,l} \log \frac{\pi_l}{k_{u,l}} + k_{u,l} \log \left[ \prod_t p_u(t)^{z_u(t)} (1 - p_u(t))^{(1-z_u(t))} \right] \right]$$

$$= \nabla_{x_l} \sum_u \sum_l \left[ k_{u,l} \log \frac{\pi_l}{k_{u,l}} + k_{u,l} \sum_t \left[ z_u(t) \log p_u(t) + (1 - z_u(t)) \log(1 - p_u(t)) \right] \right] \quad (18)$$

$$= \sum_u \sum_l \left[ k_{u,l} \sum_t \left[ z_u(t) \nabla_{x_l} \log p_u(t) + (1 - z_u(t)) \nabla_{x_l} \log(1 - p_u(t)) \right] \right]$$

$$= \sum_u \sum_l \left[ k_{u,l} \sum_t \left[ \frac{z_u(t)}{p_u(t)} \nabla_{x_l} p_u(t) - \left( \frac{1 - z_u(t)}{1 - p_u(t)} \right) \nabla_{x_l} p_u(t) \right] \right]$$

$$= \sum_u \sum_l \left[ k_{u,l} \sum_t \left[ \frac{z_u(t)}{p_u(t)} - \left( \frac{1 - z_u(t)}{1 - p_u(t)} \right) \right] \nabla_{x_l} p_u(t) \right]$$

The derivative of $p_u(t)$ considering the base model BM with respect to $x_u$ keeping other parameters fixed is

$$\nabla_{x_l} p_u(t) = \nabla_{x_l} \frac{1}{1 + exp(-[(\eta_u\ A_l(t) + \beta_l) - (x_l(t) + \epsilon)])}$$

$$= \frac{exp(-[(\eta_l\ A_l(t) + \beta_l) - (x_l(t) + \epsilon)])}{(1 + exp(-[(\eta_l\ A_l(t) + \beta_l) - (x_l(t) + \epsilon)]))^2} \nabla_{x_l}(-x_l(t) - \epsilon) \quad (19)$$

$$= -\frac{exp(-[(\eta_l\ A_l(t) + \beta_l) - (x_l(t) + \epsilon)])}{(1 + exp(-[(\eta_l\ A_l(t) + \beta_l) - (x_l(t) + \epsilon)]))^2}$$

Similarly, the gradients with respect to other parameters $\eta_l$, $\beta_l$ and $p_l$ can also be calculated and the parameter updates in the **M** step can be performed via gradient descent using the following procedure. This is a standard finite mixture model where the parameters are estimated by the Expectation Maximization (EM) framework [49]. The brief steps to obtain the parameter estimates are as follows:

## 6.5 Dataset

We used the Twitter dataset released publicly by authors in [50] which analyzed how people orient to and spread rumors in social media. As discussed in that study, adapting the existing definition to the context of breaking news stories, a rumor is defined as a "circulating story of questionable veracity, which is apparently credible but hard to verify, and produces sufficient skepticism and/or anxiety so as to motivate finding out the actual truth". The tweets from that study were collected from the streaming API relating to newsworthy events that could potentially prompt the initiation and propagation of rumours. Selected rumours were then captured

**Table 4. Statistics of the data used for simulation relating the 3 events.**

|  | Charlie Hebdo | Putin Missing | Ferguson Unrest |
|---|---|---|---|
| **Network Nodes** | 17426 | 243 | 4534 |
| **Network Edges** | 33598 | 520 | 12076 |
| **# cascades** | 74 | 11 | 53 |
| **Avg. in-degree** | 1.98 | 2.12 | 3.04 |

in the form of conversation threads. The authors used Twitter's streaming API to collect tweets in two different situations: (1) breaking news that is likely to spark multiple rumours and (2) specific rumours that are identified a-priori. They collected a total of nine events pertaining to these situations but we used the threads from three events in this paper. The twitter threads were related to the following three events:

- **Charlie Hebdo Shooting**: Two brothers forced their way into the offices of the French satirical weekly newspaper Charlie Hebdo in Paris, France killing 11 people and wounding 11 more, in January 2015.

- **Ferguson unrest**: The citizens of Ferguson in Missouri protested after the fatal shooting of an 18-year-old African American, Michael Brown, by a white police officer on August 9, 2014.

- **Putin missing**: Numerous rumors emerged in March 2015 when Russian president Vladimir Putin did not appear in public for 10 days. He spoke on the 11$^{th}$ day, denying all rumors that he had been ill or was dead.

Since the publicly released dataset only contained a sample of the threads as compared to the those used in [50], we picked these three events which had relatively larger proportion of rumor threads among all the events. The authors in the study curated the annotations for the threads as to whether they were rumors or not with the help of several journalists. We considered all the threads for the events which were tagged as rumors such that the stories related to these threads were later verified as false. Fact checking for these threads were performed by a group of annotators after these events and while the entire event might have been later described as true, there were threads related to those events that spread misinformation. The statistics of the data pertaining to these three events is provided in Table 4. For each of the events, the dataset provides us with the following segregated information modules:

1. Initialize the parameters $\Theta$, $\mathbf{P}$ and evaluate the log likelihood of the model using Eq 14.

2. **E-Step**: Evaluate the posterior probabilities using the current values of $\Theta$, $\mathbf{P}$, with Eq 16.

3. **M-Step**: Update the parameters $\Theta$, $\mathbf{P}$ with the current values of the posterior using the gradients obtained through maximization of the log likelihood with respect to parameters.

4. Evaluate the log-likelihood with the new parameter estimates. If the log-likelihood has changed by less than some small $\epsilon$, stop, else reiterate Steps 2 and 3.

**Algorithm 1**: Simulating the diffusion of cascade $q$ based on social influence and individual decisions.

```
Input: T_thresh, Activated Set A (till time T_thresh), Time limit T_sim, Θ,
P, σ, users V_q, G, Model Type MT
Output: Diffusion Node Set DF_q[t], ∀t ∈ [1, T_sim]
activated ← A
DF_q[0] ← {}
for t = 1 to T_sim do
  curr_activated ← {}
```

```
    DF_q[t] ← DF_q[t - 1]
    for each agent u∈ V_q\activated do
      l_v ← argmax_l (p_l f(z_u(T_thresh);θ_l)) / (∑_l p_l f(z_u(T_thresh);θ_l))
      /* Calculate individual factor μ_u(t)            */
      ε ~ N(0,1)
      compute μ_u(t) with Eq 6 using ε and x_{l_v}
      /* Calculate utility from influence ζ_u(t)        */
      if MT == BM then
        compute ζ_u(t) with Eq 8
      else
        compute A_u(t) using G with Eq 9
        compute ζ_u(t) with Eq 8
      /* Calculate probability of activation p_u(t)          */
      p_u(t) ← 1 / (1+exp(-[ζ_u(t)-μ_u(t)]))
      if p_u(t) > 0.5 then
        DF_q[t] ← DF_q[t] ∪ {v}
        curr_activated ← curr_activated ∪ {v}
    activated ← activated ∪ curr_activated
  return DF_q
```

1. *Who-follows-who network*: This social network is sampled to cater to specific users who participated through either replying to tweets or retweeting that specific event. This allowed us to focus our simulation for each event using this network $N$ instead of using one large social network common to all events. It helped in part by enabling us in the evaluation to compare our set of activated individuals at each time step to the actual users who were activated.

2. *Retweet cascades*: We considered retweet cascades and did not include users who simply replied to a particular tweet since it is challenging to deduce whether a user agreed to the agenda of the cascade while replying—the notion that induces a cascade of like minded individuals. So we restricted users in the cascade to those who only retweeted the source tweets.

We operationalized our simulation of behavior diffusion described in details in the next section, for each event separately. We used the follower networks for each event to simulate the diffusion and used the retweet cascades to learn the agent parameters specific to each event.

## 6.6 Simulation algorithm

We now describe the algorithm for operationalizing the simulation of behavior diffusion based on the influence setup described in Section 6.3. As mentioned before, agents refer to users in the social network and a one-to-one mapping to the actual network of the events from the dataset. We used the follower networks relevant to each event, which are directed in its edges and we refered to such networks as $G$. The algorithm for the agent-based model for social influence based diffusion process is described in Algorithm 1.

The diffusion simulation unfolds in discrete time steps and at each step, multiple agents can change their states (initial state being the state where the user/node has not reshared the message)—however, once they transition from non-shared to shared state, they cannot switch back. We observed the users who participated in the first $T_{thresh}$ steps of the cascades and used that to learn the parameters, specifically the latent class $l$ specific parameters $θ_l$, $p_l$ for all classes $l ∈ [1, L]$. We then use the activated nodes (which have already reshared the rumor prior to $T_{thresh}$) along with these learned parameters as input to the simulation algorithm. From

thereon, we ran the algorithm for a span of $T_{sim}$ discrete time steps. For each $t \in [1, T_{sim}]$, the algorithm outputs the number of agents who reshared the rumor message at time $t$ or were activated at time $t$. In each step $t$, we looped through all the agents in the network that had yet to reshare the message. Since we projected each agent into a mixture of latent classes $L$, prior to the simulation step, we needed to categorize each agent into one of the latent classes in order to compute $p_u(t)$ with the respective parameters of the latent class they belong to. We observe $D_u(t)$, $A_u(t)$ of each user $u$ till $T_{thresh}$ in the real world data and then the latent class can be decided by the maximum of posterior $\arg \max_l \; k_{u,l}$ (Eq 16) from the data. Then the probability of activation is computed based on Eqs 10 and 11 depending on whether the base model or the augmented exposures model is used (the algorithm mentions the general form of the equation based on Eq 5). If this calculated probability is higher than 0.5, the agent is activated and the simulation continues for other agents for that time step.

## 6.7 Evaluation of ABM

One of the specific goals we had while setting up the ABM was to compare the two models we proposed—the base model which relies on the magnitude of peer signals as the factor for social influence and the augmented model which allows for accommodating the sequential changes in peer signals. To this end, we compare the diffusion trajectory of the cascades from simulations based on each model and the real world data. Note that all the parameters of the models represented in Eqs 10 and 11 are learned from the data, except for the noise random variable $\eta$ that adds uncertainty to the individual utilities. We sampled $\epsilon \sim \mathcal{N}(0, 1)$ and following this, we executed 100 runs of the simulation algorithm for each cascade (thread) for each event in the data to account for this uncertainty. For learning the parameters as mentioned before, we observed the cascades for each event till time $T_{thresh}$. Since the time span of resharing actions for each cascade is different and in the absence of any normalization procedure that could be applied to decide on a single $T_{thresh}$ for all cascades, we instead observed the first 40% of each cascade (in terms of total number of reshares in the cascade) in the chronological order of reshares. This allowed us to keep the $T_{thresh}$ dynamic for each cascade while allowing the rest of the cascade to be used for ABM evaluation.

To evaluate the simulation results, for each cascade $q$, we consider the set of users $V_q \in V(\mathcal{G})$ for each network $\mathcal{G}$ relevant to the event, such that all users in $V_q$ reshared $q$ in the time span of the cascade. We ran each simulation for $T_{sim} = 20$ steps. For each time step $t$ in our simulation, we computed the following metric for each cascade: $\frac{|V_q \cap DF_q[t]|}{|V_q|}$, the number of actual activated users which are also part of the activated users from the simulation algorithm at time $t$. We call this measure the *True Diffusion Rate* of our simulation algorithm. The metric does not measure prediction results here since there is no way in which we can precisely map a time step in our simulation to a numeric time interval in the real world dataset. The metric allows us to measure recall over the users who reshared the rumors while allowing us to simulate the trajectory of the diffusion over time. Fig 12 show the results of the ABM simulation for the two simulations. For each event, we ran the 2 models as described in Section 6.3. So we have a total of six models and we learned six sets of parameters and run 100 simulations for each model that learn the parameters of the ABM.

Fig 12(a), 12(b) and 12(c) show the plots corresponding to the true diffusion rate over time and it compares the two models: the **BM** model where the peer influence at time $t$ is characterized by the number of neighbors of a user who have reshared the rumor message till $t$ and the **AEM** model where the peer influence at time $t$ is characterized by the augmented exposures till $t$ given in Eq 9. For all the results, we plotted the mean of the *True Diffusion Rate* at each

time step taking all cascades for the respective events into account. Below are the results from the Charlie Hebdo and Putin missing events followed by the Ferguson arrest events:

**Charlie Hebdo and Putin missing**: For the cascades related to both of these events, the AEM model which takes into account our notion of augmented exposures, exhibits a faster diffusion rate compared to the BM model. The results are not surprising in this scenario since the additive factor of influence coming from the spike in neighbor information in the AEM model augments the utility from social influence resulting in faster diffusion. We also observed that for the cascades in the Putin Missing events, the diffusion rate for the AEM model is an order of magnitude higher in the initial stages until seven time steps and the AEM model reaches the saturation point of the curve faster than the BM model.

On closer analysis, we found that the network structure in this case has an important role to play. The degree distribution for the network used for the Putin missing event displayed in Fig 11 shows that there are only a few nodes with high in-degree or the number of potential peer signals. So the spikes in the adoption curve for the Putin missing event happens when these few nodes with high in-degree (or higher potential of exposure to peer siganls) are exposed to the message from multiple peers and they reshare the message. This happens earlier for the AEM model than the BM model. Consequently, this result shows that faster diffusion in real world networks can often be attributed to the presence of a few nodes that are more susceptible to exposure, resulting in faster adoption at a population level. It shows how peer signals in the initial stages can drive diffusion faster for the rest of the trajectory.

We also observed that in both of these events, the dynamics of adoption do not vary much in that at no point does the adoption rate induced by the BM model surpass the AEM model. This also suggests that when population-level adoption is faster shown by the fact that in both cases, almost 70% of the actual users who reshared the message were activated in our AEM model by Time Step 7, user uncertainty does not account for much and the peer signals drive the dynamics.

**Ferguson arrest**: For the cascades belonging to the Ferguson arrest event, we fund that in contrast to the results discussed above, the AEM model resulted in slower diffusion than the base model after time step 4 as shown in Fig 12(c). On closer analysis, the reasons behind this can be attributed to two key observations: (1) Slow initial diffusion prior to the start of simulation for cascades belonging to the Ferguson arrest event. It took an average of $T_{thresh} = 25$

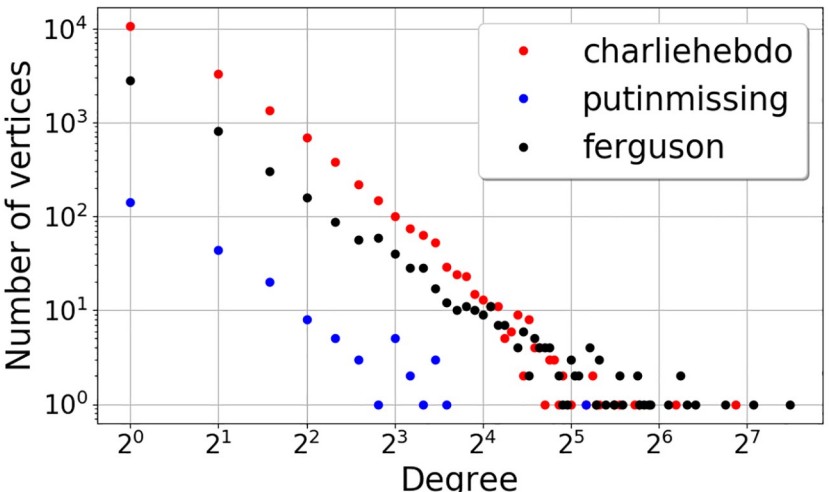

**Fig 11. Degree distribution of the follower networks.**

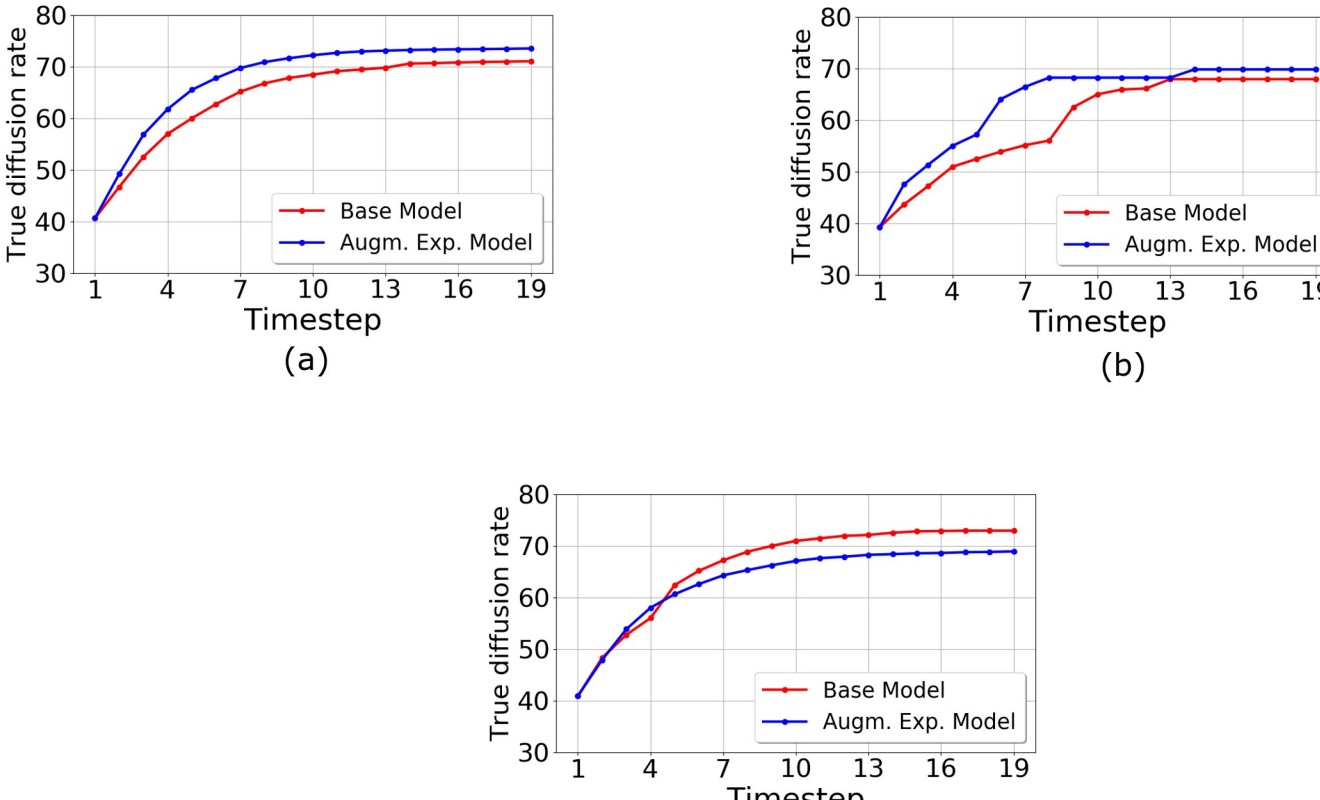

**Fig 12. ABM results for the simulation using Algorithm 1 on three Twitter who-follows-whom network for the events (a) Charlie Hebdo, (b) Putin missing and (c) Ferguson unrest.**

and 31 hours for the cascades to reach 40% of their final affected population for the Charlie Hebdo and the Putin missing events respectively, while it took roughly $T_{thresh}$ = 134 hours to reach the same 40% of the final size for the cascades in the Ferguson unrest events. This also led to the models setting high values of $\mu_u(t)$ from the learning procedure prior to start of simulation i.e. the user uncertainties were learned to be higher at the start of the simulation resulting in slower diffusion through the simulation given that it took 10 time steps to reach 70% of the population in contrast to seven time steps for the other two events. (2) The slower diffusion led to many nodes not experiencing any increase in peer signals, resulting in lower probability of activation over time, since the factor $e^{A_u(t)-A_u(t-1)}-1$ could be less than 1 when $A_u(t) = A_u(t-1)$ i.e. there was no increase in peer exposures which happens to be the case in this situation.

Consequently, the social influence factor dropped in AEM model after Time Step 4 compared to the BM model which explains the plot Fig 12(c). This suggests that in networked situations unlike our experimental environment, slower initial diffusion can result in initial higher uncertainty which can eventually result in the decay of the influence factor later on. In such cases delayed stimulus through interventions would be the only way for peer influence to play a bigger role which of course would be undesirable in such sub-optimal choice diffusion scenarios. As mentioned before, one of the limitations of this model lies in the setup that a user cannot transition back to a state it has explored. This limits us in measuring the retention effect

concurrent to what we tested for the Early Cascade phenomenon. However we believe that our binary choice model can be extended to multiple choice data given relevant case studies. Tthis would then allow for understanding of the retention effect in the real world and whether early exposures are not an effective tool for retention over the long run as we concluded from our controlled experiment setup.

## 7 General discussion

The primary goal of our studies was to examine how individuals' decisions are influenced by the decisions of others, particularly when they are exposed to different cascading peer influence patterns. Through a behavioral experiment and a data-driven agent-based model, we explored how social influence can play a role in sub-optimal behavior diffusion. Specifically, we investigated how temporal patterns of influence, by and large, affect decision-makers when the decisions have utilities. We conducted two sets of studies. In Study 1, we developed a controlled experimental setup that divided participants into five groups based on the manner in which they were treated to peer signals or the PoI over time. Our first hypothesis, that studied this effect of PoI on behavior outcome, confirmed that an early exposure to signals commands the most success in terms of desired outcome when aggregated over all of the time steps of the game. However, it did not shed much light on the temporal variations between-groups in decision making. Based on a second hypothesis that attempted to study this aspect, we analyzed the influence of the quantity of peer signals across the time steps. We found that the effect of the same quantity of signals could have a substantially different effect based on the Po. While early exposure to a large peer influence can decay very fast thus failing the retention effect one would hope would come from early exposures, a delayed stimulus in peer signals proves to be a successful strategy in resurgence of the desired outcome of influence.

The first study was conducted to focus on the nature of peer influence on sub-optimal choice diffusion. However, in real world applications, information diffuses through networked environments. Additionally, the controlled settings implicitly do not allow us to measure the role of individual uncertainty or private information which is an additional cognitive factor that remains difficult to be measured. To complement the online controlled study, we conducted a simulation on real world Twitter networks to measure the impact of influence decisions towards rumor diffusion. We find that over a long a period of time, the influence effect from majority of an individual's peers were sufficiently large to persuade users to follow their neighbors despite having to reshare a message of questionable veracity. While that is intuitive, we found that surprisingly when information diffusion is slow at the beginning of a cascade, individual uncertainty can play a substantial role as time progresses and can itself impact this influence effect and thus impacting the trajectory of adoption. Our conclusions deviate from the existing notions of networks being the main constituents controlling both the probability of successful social influence and the resharing models. Our conclusions point to adversarial situations where the network organization can be manipulated to now be used for devising successful social influence mechanisms. These strategies or patterns of influence can become the confounding factors behind the outcomes and therefore these deserve to be studied in more detail.

Such conclusions can have diverse implications in the real world, where strategies to encourage harmful decisions could be weaponized in adversarial situations. Social influence is key to technology adoption, and research on the role of persuasion in security technology adoption indicates that various social influence factors impact a user's decision when making decisions to purchase or use a given technology. However, these studies have primarily investigated the role of benign social influence and not how it can be harnessed to harm users, e.g. by

cyber-adversaries. Specifically, social influence has primarily been studied in the context of it having a net positive impact on society, especially when considering the utility of the decisions made through influence. Given the slew of recent events in which cyber warriors exploit social media with malicious intent, researchers and policy-makers are reconsidering the role of social influence as a tool for change. Together, these simulations and the behavioral experiment illustrate the power of multidisciplinary, complimentary work in the computational social sciences. Use of modeling and the principles of experimentation allow us to more holistically study the effect of social influence on decision-making.

Current research thus sheds light on the principled manner in which influence patterns can be harnessed to achieve a desired outcome that could be harmful in myriad ways. While it is evident from our growth models, that the role of the magnitude of signals on the decision outcome is significant across time, we also found that the absolute quantity of peer signals can command different probabilities of success when disseminated using different mechanisms. Following this, we see our research being extended into multiple directions: a straightforward extension would be to test these patterns at scale when the number of peers that a user is subject to is large and so the duration of the game is also proportionally extended. A second direction can be extended to situations where the users also stand to lose money for certain decisions—in the real world these could be factors like costing users their credibility and reputation for the wrong choice and it remains to be seen whether users would still be tempted to explore or if they would exploit the best option more often. Similarly, the agent-based model can be extended to multi-armed bandit situations where specific algorithms could be devised based on regret achieved from a convex combination of utilities derived from social influence and its own experience.

## Supporting information

**S1 Appendix.**
(ZIP)

## Acknowledgments

Some of the authors are supported through the ARO grant W911NF-15-1-0282 and W911NF-19-1-0066. Sandia National Laboratories is a multimission laboratory managed and operated by National Technology & Engineering Solutions of Sandia, LLC, a wholly owned subsidiary of Honeywell International Inc., for the U.S. Department of Energy's National Nuclear Security Administration under contract DE-NA0003525. This paper describes objective technical results and analysis. Any subjective views or opinions that might be expressed in the paper do not necessarily represent the views of the U.S. Department of Energy or the United States Government. We would like to thank Glory Emmanuel-Aviña and Victoria Newton for their insights and assistance with preliminary experiment design.

## Author Contributions

**Conceptualization:** Soumajyoti Sarkar, Paulo Shakarian, Kiran Lakkaraju.

**Data curation:** Soumajyoti Sarkar, Kiran Lakkaraju.

**Formal analysis:** Soumajyoti Sarkar.

**Funding acquisition:** Paulo Shakarian, Kiran Lakkaraju.

**Investigation:** Soumajyoti Sarkar, Danielle Sanchez, Mika Armenta.

**Methodology:** Soumajyoti Sarkar, Kiran Lakkaraju.

**Software:** Soumajyoti Sarkar.

**Supervision:** Paulo Shakarian, Kiran Lakkaraju.

**Validation:** Soumajyoti Sarkar, Paulo Shakarian, Danielle Sanchez, Mika Armenta, Kiran Lakkaraju.

**Visualization:** Soumajyoti Sarkar.

**Writing – original draft:** Soumajyoti Sarkar, Danielle Sanchez, Mika Armenta, Kiran Lakkaraju.

**Writing – review & editing:** Soumajyoti Sarkar, Paulo Shakarian, Danielle Sanchez, Mika Armenta, Kiran Lakkaraju.

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
