## [Decision Letter · Decision Letter 0]

9 Apr 2020

PONE-D-20-05599

Use of a controlled experiment and computational models to

measure the impact of sequential peer exposures on decision

making

PLOS ONE

Dear Mr. Sarkar,

Thank you for submitting your manuscript to PLOS ONE. After careful consideration, we feel that it has merit but does not fully meet PLOS ONE’s publication criteria as it currently stands. Therefore, we invite you to submit a revised version of the manuscript that addresses the points raised during the review process.

We recommend that it should be revised taking into account the changes requested by the reviewers. To speed the review process, the revised manuscript will only be reviewed by the Academic Editor in the next round.

We would appreciate receiving your revised manuscript by May 18 2020 11:59PM. To enhance the reproducibility of your results, we recommend that if applicable you deposit your laboratory protocols in protocols.io, where a protocol can be assigned its own identifier (DOI) such that it can be cited independently in the future. For instructions see: http://journals.plos.org/plosone/s/submission-guidelines#loc-laboratory-protocols

We look forward to receiving your revised manuscript.

Kind regards,

Baogui Xin, Ph.D.

Academic Editor

PLOS ONE

Journal Requirements:

1. Please provide additional details regarding participant consent. In the ethics statement in the Methods and online submission information, please ensure that you have specified (1) whether consent was informed and (2) what type you obtained (for instance, written or verbal, and if verbal, how it was documented and witnessed). If your study included minors, state whether you obtained consent from parents or guardians. If the need for consent was waived by the ethics committee, please include this information.

2. Please provide additional information about the participant recruitment method and the demographic details of your participants. Please consider adding such information as a) the recruitment date range (month and year), b) a description of any inclusion/exclusion criteria that were applied to participant recruitment, c) a table of relevant demographic details, d) a statement as to whether your sample can be considered representative of a larger population, e) a description of how participants were recruited, and f) descriptions of where participants were recruited and where the research took place.

3. Thank you for including your competing interests statement; "The authors have declared that no competing interests exist."

We note that one or more of the authors are employed by a commercial company: Sandia National Laboratories

3. Thank you for providing the following data availability statement:

“All data from our online controlled experiments with human participants cannot be released due to ethical limitations.

The data for our simulation of diffusion has been obtained from a publicly available dataset in https://journals.plos.org/plosone/article?id=10.1371/journal.pone.0150989”

Please note, PLOS journals require authors to make all data necessary to replicate their study’s findings publicly available without restriction at the time of publication. When specific legal or ethical restrictions prohibit public sharing of a data set, authors must indicate how others may obtain access to the data.

For more see: https://journals.plos.org/plosone/s/data-availability

Before we proceed, please address the following points:

a. Please clarify if you are able to share anonymized data for all figures and tables in your manuscript. If there are no restrictions, please upload the minimal (anonymized) data set necessary to replicate your study findings as Supporting Information files or to a stable, public repository and provide us with the relevant URLs, DOIs, or accession numbers. For a list of recommended repositories, please see our Data Availability page (http://journals.plos.org/plosone/s/data-availability#loc-recommended-repositories). You also have the option of uploading the data as Supporting Information files, but we would recommend depositing data directly to a data repository if possible.

b. If there are acceptable restrictions in place on the public sharing of the data underlying your study, please kindly clarify in detail the reasons for data restriction (e.g., data contain potentially sensitive information, data are owned by a third-party organization, etc.) and who has imposed them (e.g., an ethics committee).

c. Please also provide a non-author point of contact (e.g., data access committee, ethics committee, or other institutional body) where data requests may be made. Note that it is not acceptable for the authors to be the sole named individuals responsible for ensuring data access. If data are owned by a third party, please indicate how others may request data access.

Thank you for your assistance. With the information you provide we can update your data availability statement on your behalf.

Reviewers' comments:

Reviewer's Responses to Questions

**Comments to the Author**

1. Is the manuscript technically sound, and do the data support the conclusions?

Reviewer #1: Partly

Reviewer #2: Partly

Reviewer #3: Yes

2. Has the statistical analysis been performed appropriately and rigorously? 

Reviewer #1: I Don't Know

Reviewer #2: Yes

Reviewer #3: No

3. Have the authors made all data underlying the findings in their manuscript fully available?

Reviewer #1: No

Reviewer #2: No

Reviewer #3: No

4. Is the manuscript presented in an intelligible fashion and written in standard English?

Reviewer #1: Yes

Reviewer #2: Yes

Reviewer #3: Yes

5. Review Comments to the Author

Reviewer #1: The paper is trying to make a point that sequential exposure of information pattern has important impact on the human decision. My major concern is that the justification of selecting the specific controlled experiment and the whole procedure of the experiment is not clear. Will the conclusion still holds if applied to other games? So the finding seems not very convincing.

Reviewer #2: Use of a controlled experiment and computational models to measure the impact of sequential peer exposures on decision making

PONE-D-20-05599

This is an interesting study with a good design of the experiment. The paper focuses on the social influence of peers on human decision-making. It is especially valuable the effort to test the sequential impact of others in the decision-making of participants, and how this sequence evolves over time (uniform; linear cascade; delayed cascade; and early cascade). However, I have a number of concerns that are described below.

Regarding the literature that the authors reviewed, I miss some important previous efforts that are very connected to the research study. It is very recommendable that the authors establish the relation between this literature (see below) and their study, especially to delimitate the specific contribution of the paper.

-Literature about the dual-process theory and heuristics in how signals of the environment influence the decision-making (e.g., Kahneman, 2003; Sanjari, Jahn, & Boztug, 2017). Signals from others could have a heuristic role, limiting a deliberative or optimal decision-making.

-Literature related to the influence of the group on the decision-making of the individual (conformity). The classical experiments conducted by Asch (1956) demonstrated the influence of the group on bad decision-making. In addition, the decision-making was analyzed sequentially, as it is in the present paper. Other examples exist for the use of technology (e.g., Hertz & Wiese, 2018).

I agree with the idea that the main contribution of this study is the consideration of the sequence in the decision-making. The manuscript will improve very much if the introduction and the rationale of hypotheses focus much more on the evolution of decision-making based on the signals that come from others. The authors have different types of possible temporal evolution (uniform; linear cascade; delayed cascade; and early cascade) that could be anticipated and described in more detail before the methodology section. The authors have the opportunity to propose and test mechanisms of temporal evolution that depend on how other actors in the environment behave. Because this could be the relevant contribution of the paper, it is necessary that “time” (connected with the actions of others) has a more prominent role. Accordingly, I also strongly suggest to reword the hypotheses.

I think that the experiment will improve very much if, congruently with the importance of “time” I stressed above, the authors add a computation of latent growth curves (Bliese & Ployhart, 2002; Pitariu & Ployhart, 2010), or other similar analysis, to explore the form of the decision-making evolution (lineal, quadratic, cubic). This also permits to check the change in the decision-making depending on the change in the behavior of others.

I will appreciate more information about participants. Could you please provide more information about the characteristics of participants, demographic information, etc.? Were there people who declined to participate? I know there is a test for biases in the Appendix, but if there were people who declined to participate, has their influence been controlled in any way? Why there is a difference in the size of the groups? More clarification is needed.

I miss much more elaboration in the “Conclusions” section (maybe “Discussion”). I will appreciate it if the authors can extend it considering the importance of time and evolution of decision-making according to the behavior of others.

Please, review typos (e.g., “the6mber” on page 7).

Ash, S. (1956). Studies of independence and conformity: A minority of one against a unanimous majority. Psychological Monographs, 70

Bliese, P. D., & Ployhart, R. E. (2002). Growth modeling using random coefficient models: model building, testing, and illustrations. Organizational Research Methods, 5, 362-387.

Hertz, N., & Wiese, E. (2018). Under Pressure: Examining Social Conformity With Computer and Robot Groups. Human Factors, 60, 1207–1218.

Kahneman, D. (2003). A perspective on judgment and choice: mapping bounded rationality. The American psychologist, 58 9, 697-720.

Pitariu, A. H., & Ployhart, R. E. (2010). Explaining change: theorizing and testing dynamic mediated longitudinal relationships. Journal of Management, 36, 405-429.

Sanjari, S. S., Jahn, S., & Boztug, Y. (2017). Dual-process theory and consumer response to front-of-package nutrition label formats. Nutrition Reviews, 75, 871–882.

Reviewer #3: The key contributions and research questions are to understand how patterns of influence impact adoption decisions and how peer influence impacts suboptimal decision making. This is a very important and timely topic of study. The RCT and ABM both provide unique insight to how information signals influence decision making. The two types of decisions, one and adoption decision and the other a decision to share information, seem like distinctly different types of decision-making. It would be helpful if the authors spent more time explaining the link between the two experiments. This almost feels like it could be two different papers, one on the RCT and one on the ABM. Please carefully proofread the document. There are many complex sentences with grammatical or typo errors that make it difficult to read, including in the abstract.

My review focuses mainly on the RCT as this is my expertise. I am only familiar with ABM, but not an expert, and will note that I don’t think the ABM portion is very accessible as written. More clarity and definition of terms would be helpful. Additionally, the authors should address limitations of both studies, as well as the limitations in comparing results across experiments.

The paper would benefit from a stronger conclusion with a more succinct overarching takeaway, rather than a nuanced finding, and in layman’s terms. In general, the conclusion falls flat compared to the discussion and analysis sections. Using layman’s terms and including clear statements, in layman’s terms, of the highlighted key findings and the real-world implications of those findings would make the study more accessible and relevant to non-experts.

General:

Abstract. Last sentence is unclear.

End of intro, 4 bullet points on findings. The sentences are long and a bit hard to follow. This would be more impactful, especially to non-expert readers if they were more succinct summary statements, followed by more in-depth discussion. Generally, the introduction doesn’t clearly lay out the study. The paragraph from lines 68-95 begins to address this, but gets into more detail than is needed for the intro without addressing the main structure and reasoning behind the different pieces of the work, e.g., a framework and AMT game are mentioned, but the ABM doesn’t come up until bullet three of findings.

The number of groups, variables, and timing create a fairly complex set of experimental conditions. A schematic or flow chart of the experiment would be very helpful to understand the conditions and timing. Additionally, section 4.1 has some redundancy and circling back around to the same points makes it seem more complex and harder to follow. A more linear explanation of the parts would improve clarity.

Section 4.1. In lines 311-312, are the signaler and peers referred to bots? In line 314, peers are signified as (bots). If these are the same peers from 312, please indicate at first reference. Line 316, please explain “do not share any topology”.

Please state explicitly if incentives were provided in all 18 stages.

How was the incentive level selected? $0.02 seems low to motivate optimality.

Section 5.1 states that EC and LC “prevent more attacks.” However, the table column labeled “Average number of attacks prevented” has the lowest number for EC and LC rows. Table 1 description should clarify “The lower attacks” as “lower number of attacks prevented”. Please include whether the difference in attacks prevented is statistically significant in Table 1.

Section 5.2: Line 379 states suboptimal technology C, how does the letter relate to provider or decision numbers? Starts using the terminology “received decision”; what does that mean? In line 382, what “ratio”. Not sure what “highest” indicates; highest what? Why are the different groups “sent” a different decision?

Meaning of figure 5 is unclear. An example of how to read this would be helpful.

Line 396 states 2-sampled t-test. Why wasn’t an ANOVA performed first followed by pair-wise tests when the ANOVA result was significant? ANOVA is standard practice for multiple treatment groups.

Line 417. Please show statistical results for both NM and UM consistently. Also, a reminder of what “decision 5” means and why it matters would help with following the results.

Please use consistent language across all three results, e.g. decision 5, optimal choice, technology C, etc. These results are difficult to follow and compare. A table showing the results of the tests for each group and relationship to Hypotheses would be helpful.

Section 5.3. This is hard to follow. Section 5.2 indicates that the EC pattern has a different impact than the other PoIs. The paragraph starting at line 461 also indicates differences, but 5.3 contradicts this. Line 476 indicates a different experimental setup for influencing the peer social signal versus one suboptimal choice. The experimental description did not make it clear that there were two setups of this nature.

Section 5.4. Line 489. The “e.g.” example doesn’t really clarify the statement since it doesn’t compare two groups.

Figure 8. Is the number of influencers or the number of influence signals? Consistent terminology is key for clarity given the complexity of the experiment. s is defined as signal quantity in the definition line 493.

Section 5.5. A comparison with how the outcome between EC and DC supports, confirms, relates to, etc. the EC v DC comparison in section 5.4 would be helpful, especially for those of us with limited time to work through the implications.

Section 6. It would be useful to link the ABM to the RCT more tightly. As it stands, the tie between the two experiments isn’t strong enough. This seems like two different studies, tangentially related, crammed into one paper. Since the target behavior has changed, i.e., selection of tech with clear benefit/performance v sharing info with no clear benefit, it’s not obvious that results of the RCT are applicable to the premise of the ABM. The use of “optimal” decision for sharing information with no clear utility, seems like a stretch. Here optimal seems like a stand-in for “factual.”

Section 6.1 As stated, line 620-623 indicates that the same data was used for training and validating, which is self-reinforcing. Please clarify. Lines 623-6 are unclear.

Section 6. I’m familiar with ABMs, but not an expert. This section was hard to follow. The results are stated as having two results, but there seem to be a number of results in each bullet point. The significance/implications of the results are not obvious for the layperson and requires more development and explanation.

Appendix. Including the survey and definitions of variables would be helpful. Were the significances tested in Appendix A corrected for multiple-comparisons? There seem to be a lot of variables tested.

Appendix B is self-referential.

Minor

More definitions would be helpful for the reader, e.g., opacity problem line 95

Line 134. The paper organization skips section 2.

Line 154 consider referencing Beck, Lakkaraju & Rai 2017 on info frequency.

Line 346 refers to its own section.

Section 6.3.1. Line 685 has a “2)” but there isn’t a 1)

Section 6.3.3. Equation 4 introduces a sub-u (t), but doesn’t define it.

6. PLOS authors have the option to publish the peer review history of their article (what does this mean?). If published, this will include your full peer review and any attached files.

Reviewer #1: No

Reviewer #2: No

Reviewer #3: No

---

## [Author Response · Author response to Decision Letter 0]

2 Jun 2020

Dear Editor and the Anonymous reviewers,

We take this opportunity to first thank all the reviewers for all their insights that would help make this paper more readable and improve the clarity of the contributions from this research investigation. We have sincerely gone through all the comments from the reviewers. Please find the responses to each question separately with the modifications and their justifications addressed to each comment, in the new revised draft. 

We have addressed the comments of the 3 reviewers separately (with no specific priority - just in order that we received). All modifications in the original pdf have been marked in red. To facilitate for faster reviews, some of the modifications of the pdf have been copied here in response to the questions which demanded them. These parts have been marked in blue in this document.

Review 1: 

The paper is trying to make a point that sequential exposure of information pattern has important impact on the human decision. My major concern is that the justification of selecting the specific controlled experiment and the whole procedure of the experiment is not clear. Will the conclusion still holds if applied to other games? So the finding seems not very convincing.

 Response: This specific controlled experiment and the two hypotheses that we present in this paper builds up on the findings of a few papers trying to study social influence when the magnitude of peer signals are considered important - see papers 13, 17, 35. Additionally we have reworded and developed the hypotheses through the revisions, see later points in this rebuttal for the edits. What remained to be seen apart from the findings in those papers is the understanding aspect of whether the manner in which a user sees its peer adopting decisions, can alter the previous findings. Within a networked environment, such cascading influence patterns as we studied would be hard to control and observe at scale - hence the game setup with our controlled settings. With networked environments, we need to modify the game so as to be able to observe these influence patterns. But would the conclusions hold true in a networked environment - this is where we specifically try to see whether a simple rule derived based on the experimental setup could reveal something interesting in real world case studies - and this is where we studied the rumor diffusion ABM model comparing the signal magnitude as a base measure.

Review 3:

The paper would benefit from a stronger conclusion with a more succinct overarching takeaway, rather than a nuanced finding, and in layman’s terms. In general, the conclusion falls flat compared to the discussion and analysis sections. Using layman’s terms and including clear statements, in layman’s terms, of the highlighted key findings and the real-world implications of those findings would make the study more accessible and relevant to non-experts

Response: We have reworked on the Conclusion, now renamed as Section 7: General Discussion on Page 29 and we have summarized the results and the implications from our studies especially, in relation to previous works and its future extensions. We have tried to be as specific about the 3-4 main conclusions we have derived from both the studies to avoid overarching paragraphs. Please refer to this section.

My review focuses mainly on the RCT as this is my expertise. I am only familiar with ABM, but not an expert, and will note that I don’t think the ABM portion is very accessible as written. More clarity and definition of terms would be helpful. Additionally, the authors should address limitations of both studies, as well as the limitations in comparing results across experiments.

 Response: It would have helped if the reviewer could have provided a few examples of areas to improve - anyway, we have changed quite a bit of the ABM work starting Section 6 - the connection to the controlled experiment and motivations on Page 17 and the conclusions from the ABM in more details on Pages 26-27. We made several changes to the ABM section to make it more readable and additionally, we have included a notations table on Page 7 and made the symbols consistent throughout the paper for better readability. 

Abstract: Last sentence in unclear

Response: We realize that the abstract had not been written in the best manner to capture the conclusions from our behavioral study and we have reframed the last few lines of the pdf. The extract from the pdf now reads

“To better understand how these rules of peer influence could be used in modeling applications of real world diffusion, we use our behavioral findings to simulate spreading dynamics in real world case studies. We specifically try to trade-off cumulative influence effect and user uncertainty in adoption and measure its outcome on rumor diffusion, which we model as an example of sub-optimal choice diffusion. Together, our simulation results indicate that sequential peer effects from the influence decision overcomes individual uncertainty to guide faster rumor diffusion over time. However, when individuals observe peer signals late in the process, user uncertainty from the beginning can have a substantial role in deciding the adoption trajectory of a piece of questionable information.”

End of intro, 4 bullet points on findings. The sentences are long and a bit hard to follow. This would be more impactful, especially to non-expert readers if they were more succinct summary statements, followed by more in-depth discussion. Generally, the introduction doesn’t clearly lay out the study. The paragraph from lines 68-95 begins to address this, but gets into more detail than is needed for the intro without addressing the main structure and reasoning behind the different pieces of the work, e.g., a framework and AMT game are mentioned, but the ABM doesn’t come up until bullet three of findings.

 Response: We have changed the introduction considerably keeping this mind and have added the introduction to the 2 components of our work - the online experimental study and the data-driven model on a separate observational data. Please refer to the modified sections in the original pdf marked in red in the Introductions section. Similarly, the bullet points at the end have been simplified to be less technical and to present a more high-level overview of our findings.

The number of groups, variables, and timing create a fairly complex set of experimental conditions. A schematic or flow chart of the experiment would be very helpful to understand the conditions and timing. Additionally, section 4.1 has some redundancy and circling back around to the same points makes it seem more complex and harder to follow. A more linear explanation of the parts would improve clarity.

Response: We do agree that there were some issues with the flow of the design of the experiments in Section 4 and Section 4.1 in the previous draft. We have modified the flow and have ensured no redundancy of circling back of the content. Please refer to the modified sections in the original pdf marked in red in these sections.

Section 4.1. In lines 311-312, are the signaler and peers referred to bots? In line 314, peers are signified as (bots). If these are the same peers from 312, please indicate at first reference. Line 316, please explain “do not share any topology”.

Response: We have modified Section 4.1 to ensure that the reader has the understanding that the peers are actually bots and not subjects recruited in AMT. The first instance where this is mentioned in in Line 313 which reads:

“In the second phase of the experiment which started at time step 13, we introduced interventions by allowing participants access to extra information from 6 other individuals which are supposedly their peers (and which are bots controlled by us)”

We have removed all mentions of bots in the paper and only the description in Figure 4 has a mention of it to emphasize that it is the control mechanism for influence. We have removed the phrase “do not share any topology” and have replaced it in Line 331 with 

“Note that we attempt to avoid network effects by randomizing this technology or the \\textit{influence decision} $C_u$ specific to the user $u$ - this allows us to avoid any deliberate collisions among peer choices of different users that could be representative of network effects in the real world”

Meaning of figure 5 is unclear. An example of how to read this would be helpful.

Response: We have updated the description of Figure 5 - to improve its readability, we introduced a notation for the measure described in Section 5.2 starting Line 408. 

Line 396 states 2-sampled t-test. Why wasn’t an ANOVA performed first followed by pairwise tests when the ANOVA result was significant? ANOVA is standard practice for multiple treatment groups.

 Response: Since the comparisons were made separately between the 2 control groups and each treatment group, we initially did not perform a single ANOVA test that would not realize the tests that we wanted to conduct. However, upon the suggestions of the reviewer, we decided to include a basic one-way ANOVA test prior to the t-tests to give the reader about a better understanding of the significance results. This study is now included in Section 5.2 starting Line 432.

Line 417. Please show statistical results for both NM and UM consistently. Also, a reminder of what “decision 5” means and why it matters would help with following the results.

 Reponse: We do agree there were some mistakes with some of the statistics figures from the tests and their conclusions - we sincerely apologize for the confusion this might have caused - however our general conclusions from the t-tests still hold true - the ones that needed correction have been modified in Page 12 and the Appendix C tables have also been corrected for.

Please use consistent language across all three results, e.g. decision 5, optimal choice, technology C, etc. These results are difficult to follow and compare. A table showing the results of the tests for each group and relationship to Hypotheses would be helpful.

 Response: We have modified the flow of section 4.1 that provides for a more uncomplicated understanding of the design experiment. We have added following line at Line 408 in Section 5.2 that clarifies this :

To simplify nomenclature hereon, we denote the 6 available technology choices as \\textit{decision} $d_i$, $i \\in [1, 6]$. In our experimental setup and for this work throughout, we would refer to $d_1$ as the influence decision - it is the optimal choice preventing 7 attacks, while the rest of the technologies prevented 5 attacks and are being termed as sub-optimal choices.

Section 5.3. This is hard to follow. Section 5.2 indicates that the EC pattern has a different impact than the other PoIs. The paragraph starting at line 461 also indicates differences, but 5.3 contradicts this. Line 476 indicates a different experimental setup for influencing the peer social signal versus one suboptimal choice. The experimental description did not make it clear that there were two setups of this nature.

Response: Although we did not understand the exact nature of clarification requested in this comment, we clarify here that there is only one experimental setup in the paper. We felt that Line 476 did not contribute any additional information to the findings discussed in Section 5.3 so we have remove that excerpt of Line 476 from the revised paper for better clarity.

Section 5.4. Line 489. The “e.g.” example doesn’t really clarify the statement since it doesn’t compare two groups.

This has been fixed 

Figure 8. Is the number of influencers or the number of influence signals? Consistent terminology is key for clarity given the complexity of the experiment. s is defined as signal quantity in the definition line 493.

Response: Fixed to number of influence signals.

Section 5.5. A comparison with how the outcome between EC and DC supports, confirms, relates to, etc. the EC v DC comparison in section 5.4 would be helpful, especially for those of us with limited time to work through the implications.

Response: We have added some more details about the objective and conclusions in this section. Please refer to the modifications made in Section 5.6

Section 6. It would be useful to link the ABM to the RCT more tightly. As it stands, the tie between the two experiments isn’t strong enough. This seems like two different studies, tangentially related, crammed into one paper. Since the target behavior has changed, i.e., selection of tech with clear benefit/performance v sharing info with no clear benefit, it’s not obvious that results of the RCT are applicable to the premise of the ABM. The use of “optimal” decision for sharing information with no clear utility, seems like a stretch. Here optimal seems like a stand-in for “factual.”

 Response: We have added some details about this study in the Introductions section. Furthermore, we have added some more details in Section 6 to address why adding a data-driven study that extends some of the rules from our experimental study makes for an appealing real world application of our controlled setup.

Please refer to Page 17 and Section 6 for the modifications

Section 6. I’m familiar with ABMs, but not an expert. This section was hard to follow. The results are stated as having two results, but there seem to be a number of results in each bullet point. The significance/implications of the results are not obvious for the layperson and requires more development and explanation.

 Response: We agree that the conclusions from the ABM were straightway too technical for the general audience to draw any conclusions. Subsequently, we have reworded and developed a more broad understanding of the results from the simulation and how the networked settings in the simulation lead to newer results that could not be observed in the experimental settings.

Please see Page 27-28 starting Line 1045 for the changes made in response to this question.

Section 6.1 As stated, line 620-623 indicates that the same data was used for training and validating, which is self-reinforcing. Please clarify. Lines 623-6 are unclear.

This has been fixed

More definitions would be helpful for the reader, e.g., opacity problem line 95

Fixed 

Line 134. The paper organization skips section 2.

Fixed 

Line 154 consider referencing Beck, Lakkaraju & Rai 2017 on info frequency. 

Done - Now Line 178

Line 346 refers to its own section.

Fixed

Section 6.3.1. Line 685 has a “2)” but there isn’t a 1)

Fixed

Section 6.3.3. Equation 4 introduces a sub-u (t), but doesn’t define it.

 This is just the same measure - we have reordered the equality notations - this is now Equation 8.

Review 2

-Literature about the dual-process theory and heuristics in how signals of the environment influence the decision-making (e.g., Kahneman, 2003; Sanjari, Jahn, & Boztug, 2017). Signals from others could have a heuristic role, limiting a deliberative or optimal decision-making.

Response: We have included a brief connection of where social influence fits in the realm of dual influence theory in the Related Work Section on Page 5. The excerpt from the draft is copied below:

These manners of decision making have also been linked to the concept of dual process theory - the notion that two different systems of thought co-exist; a quick, automatic, associative, and affective-based form of reasoning and a slow, thoughtful, deliberative process. Fast thinking involves conditions of ``cognitive ease" and so social influence factors into this process of slowing down the decision making system by presenting alternating evidences for reconsideration.

-Literature related to the influence of the group on the decision-making of the individual (conformity). The classical experiments conducted by Asch (1956) demonstrated the influence of the group on bad decision-making. In addition, the decision-making was analyzed sequentially, as it is in the present paper. Other examples exist for the use of technology (e.g., Hertz & Wiese, 2018).

Response: We acknowledge the concerns raised by the reviewer regarding the origins of the thoughts of social influence - however we tend to disagree a bit here. While we are aware of Asch’s work on influence, there has been substantial work in recent years which have developed upon Asch’s ideas and built experiments similar to ours. As such we feel that it is appropriate to refer to more recent works in the area which are built upon Asch’s work rather than directly citing Asch’s paper which dates back a long time ago and influence related literature has evolved quite a bit since then. Our related work cites several papers on such sequential decision making scenarios and influence related to our setup.

I agree with the idea that the main contribution of this study is the consideration of the sequence in the decision-making. The manuscript will improve very much if the introduction and the rationale of hypotheses focus much more on the evolution of decision-making based on the signals that come from others. The authors have different types of possible temporal evolution (uniform; linear cascade; delayed cascade; and early cascade) that could be anticipated and described in more detail before the methodology section. The authors have the opportunity to propose and test mechanisms of temporal evolution that depend on how other actors in the environment behave. Because this could be the relevant contribution of the paper, it is necessary that “time” (connected with the actions of others) has a more prominent role. Accordingly, I also strongly suggest to reword the hypotheses.

 Response: We have reworded the hypothesis following the suggestions by this point and have also included the intuition behind selecting the hypotheses for study in this paper.

I think that the experiment will improve very much if, congruently with the importance of “time” I stressed above, the authors add a computation of latent growth curves (Bliese & Ployhart, 2002; Pitariu & Ployhart, 2010), or other similar analysis, to explore the form of the decision-making evolution (lineal, quadratic, cubic). This also permits to check the change in the decision-making depending on the change in the behavior of others.

 Response: We have included a study on growth modeling in the paper to quantify the aspect of time and signals on the outcomes. Please refer to Section 5.5. 

I will appreciate more information about participants. Could you please provide more information about the characteristics of participants, demographic information, etc.? Were there people who declined to participate? I know there is a test for biases in the Appendix, but if there were people who declined to participate, has their influence been controlled in any way? Why there is a difference in the size of the groups? More clarification is needed.

 Response: We have included data about the characteristics of the participants in the paper as well as the appendix Please refer to Section 4.2 in main paper and Appendix A in the supplementary file.

I miss much more elaboration in the “Conclusions” section (maybe “Discussion”). I will appreciate it if the authors can extend it considering the importance of time and evolution of decision-making according to the behavior of others.

We have reworded the conclusions section to include a general discussion about our studies and where it fits with real world applications.

---

## [Editor Report · Decision Letter 1]

4 Jun 2020

Use of a controlled experiment and computational models to

measure the impact of sequential peer exposures on decision

making

PONE-D-20-05599R1

Dear Dr. Sarkar,

We’re pleased to inform you that your manuscript has been judged scientifically suitable for publication and will be formally accepted for publication once it meets all outstanding technical requirements.

Kind regards,

Baogui Xin, Ph.D.

Academic Editor

PLOS ONE
---

## [Editor Report · Acceptance letter]

25 Jun 2020

PONE-D-20-05599R1 

Use of a controlled experiment and computational models to
measure the impact of sequential peer exposures on decision
making 

Dear Dr. Sarkar:

I'm pleased to inform you that your manuscript has been deemed suitable for publication in PLOS ONE. Congratulations! Your manuscript is now with our production department. 

Kind regards, 

on behalf of

Professor Baogui Xin 

Academic Editor

PLOS ONE